# On the Benefits of Attribute-Driven Graph Domain adaptation

**Ruiyi Fang**[1,*]  **Bingheng Li**[2,*]  **Zhao Kang**[4]  **Qiuhao Zeng**[1]  **Nima Hosseini Dashtbayaz**[1]
**Ruizhi Pu**[1†]  **Boyu Wang**[1,3]  **Charles Ling**[1,3]
[1] Western University  [2] Michigan State University  [3] Vector Institute
[4] University of Electronic Science and Technology of China
{rfang32,rpu2,qzeng53,charles.ling}@uwo.ca
libinghe@msu.edu   zkang@uestc.edu.cn   bwang@csd.uwo.ca

## Abstract

Graph Domain Adaptation (GDA) addresses a pressing challenge in cross-network learning, particularly pertinent due to the absence of labeled data in real-world graph datasets. Recent studies attempted to learn domain invariant representations by eliminating structural shifts between graphs. In this work, we show that existing methodologies have overlooked the significance of the graph node attribute, a pivotal factor for graph domain alignment. Specifically, we first reveal the impact of node attributes for GDA by theoretically proving that in addition to the graph structural divergence between the domains, the node attribute discrepancy also plays a critical role in GDA. Moreover, we also empirically show that the attribute shift is more substantial than the topology shift, which further underscore the importance of node attribute alignment in GDA. Inspired by this finding, a novel cross-channel module is developed to fuse and align both views between the source and target graphs for GDA. Experimental results on a variety of benchmark verify the effectiveness of our method.

## 1 Introduction

In the area of widespread internet data collection, graph vertices are frequently associated with content information, referred to as node attributes within basic graph data. Such graph data can be widely used in prevalent real-world applications, with data suffering from label scarcity problems in annotating complex structured data is both expensive and difficult (Xu et al., 2022). To solve such a challenge, transferring abundant labeling knowledge from task-related graphs is a method considered (Chen et al., 2019). Giving labeled graphs as a source to solve unlabeled graph targets has been proposed as graph domain adaptation (GDA) as a paradigm to effectively transfer knowledge across graphs by addressing distribution shifts (Shi et al., 2024).

Early works on GDA apply deep domain adaptation (DA) techniques directly, thereby (Shen et al., 2020b; Shui et al., 2023; Wu et al., 2020; Shen et al., 2020a; Dai et al., 2022) without considering the topological structures of graphs for domain alignment. To address this issue, several recent works have been proposed to leverage the inherent properties of graph topology (e.g., adjacency matrix). While these methods (Yan & Wang, 2020; Shi et al., 2023; Shen et al., 2023; Wu et al., 2023) have achieved substantial improvements by alleviating the topological discrepancy between domains, they overlook the importance of node attributes, a fundamental aspect of GDA. To verify our argument, we investigate the projected feature values[1] of graph topology and attribute on

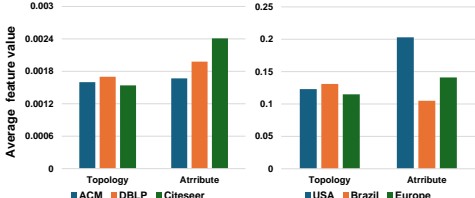

Figure 1: This represents feature value in two groups of datasets. This shows the feature value distribution gap in the attribute is larger than in the topology.

---

*Equal contribution

†Corresponding author

[1]Details on the construction of project features are presented in Section B of Appendix.

two GDA benchmarks, as shown in Figure 1. It can be observed that feature value discrepancy exists in all GDA benchmark inside datasets, with feature value discrepancy for attributes significantly larger than topology feature value discrepancy. Based on this observation, we can conclude that (1) graph distribution shift exists in both attribute and topology; (2) attribute divergence between the source and target graphs is more significant than the topology divergence.

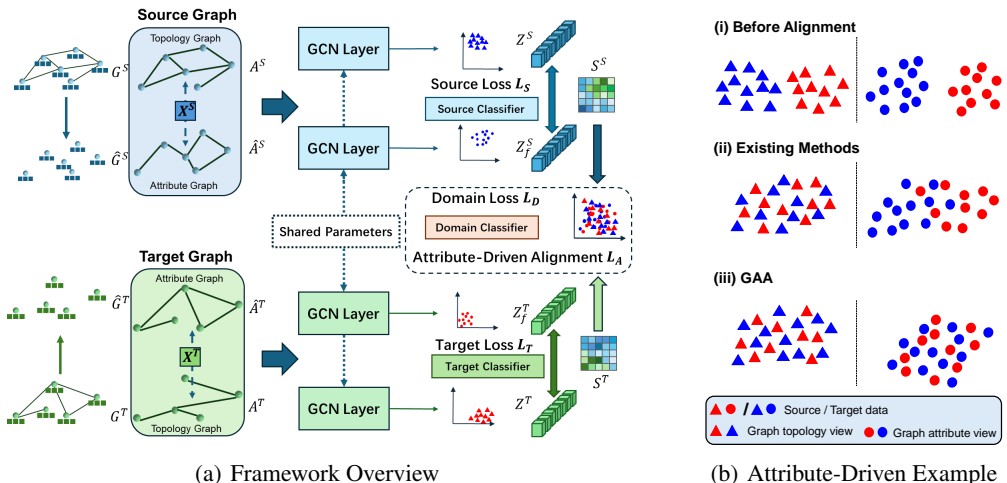

(a) Framework Overview

(b) Attribute-Driven Example

Figure 2: (a) An overview of our method. GAA gives attribute and topology graph representation, where minimizing source and target distribution shift through two views. (b)(i) Distribution shifts exist in both topology and attribute views before alignment. (ii) Existing GDA algorithms can only address graph topology shifts but not attribute shifts. (iii) GAA can address GDA attribute shifts.

Motivated by this observation, we theoretically investigate the domain discrepancy between two graphs, revealing the role of node attribute for GDA. Specifically, by leveraging the PAC-Bayes framework, we derive a generalization bound of GDA, which unveils how graph structure and node attributes jointly affect the expected risk of the target graph. Moreover, we also show that the discrepancy between the source and target graphs can be upper bounded in terms of both node attributes and topological structure. In other words, our theoretical analysis reveals that both attribute and topology views should be considered for GDA, with the former having a more significant impact on domain alignment, as shown in Figure .1.

Our theoretical insights highlight the significance of characterizing the cross-network domain shifts in both node attributes and topology. To this end, we propose a novel cross-channel graph attribute-driven alignment (GAA) algorithm for cross-network node classification, as shown in Figure.2 (a). Unlike existing methods that rely solely on topology, GAA also constructs an attribute graph (feature graph) to mitigate domain discrepancies. Furthermore, GAA also introduces a cross-view similarity matrix, which acts as a filter to enhance and integrate feature information within each domain, facilitating synergistic refinement of both attribute and topology views for GDA. Figure 2 (b) illustrates the benefits of GAA for GDA, which alleviates both attribute and topology shifts.

Our main contributions are summarized as follows:

- We reveal the importance of node attributes in GDA from both empirical and theoretical aspects.

- Motivated by our theoretical analysis, we proposed GAA, a novel GDA algorithm that minimizes both attribute and topology distribution shifts based on intrinsic graph property.

- Comprehensive experiments on benchmarks show the superior performance of our method compared to other state-of-the-art methods for real-world datasets of the cross-network node classification tasks.

## 2 RELATED WORK

Unsupervised domain adaptation is a wildly used setting of transfer learning methods that aims to minimize the discrepancy between the source and target domains. To solve cross-domain classification tasks, these methods are based on deep feature representation (Zhu et al., 2022; Zeng et al., 2024b;a), which maps different domains into a common feature space. Some recent studies have overcome the imbalance of domains and the label distribution shift of classes to transfer model well (Zeng et al., 2023; Jing et al., 2021; Zeng et al., 2025; Xu et al., 2023; Pu et al., 2025; Zeng et al., 2024c). Some novel settings in domain adaption have also gotten a lot of attention, like source free domain adaption(SFDA) (Yang et al., 2021; Xu et al., 2025), test time domain adaption(TTDA) (Wang et al., 2022). As for graph-structured data, several studies have been proposed for cross-graph knowledge transfer via GDA setting methods (Shen & Chung, 2019; Dai et al., 2022; Shi et al., 2024). ACDNE (Shen et al., 2020a) adopt k-hop PPMI matrix to capture high-order proximity as global consistency for source information on graphs. CDNE (Shen et al., 2020b) learning cross-network embedding from source and target data to minimize the maximum mean discrepancy (MMD) directly. GraphAE (Yan & Wang, 2020) analyzes node degree distribution shift in domain discrepancy and solves it by aligning message-passing routers. DM-GNN (Shen et al., 2023) proposes a method to propagate node label information by combining its own and neighbors' edge structure. UDAGCN (Wu et al., 2020) develops a dual graph convolutional network by jointly capturing knowledge from local and global levels to adapt it by adversarial training. ASN (Zhang et al., 2021) separates domain-specific and domain-invariant variables by designing a private en-coder and uses the domain-specific features in the network to extract the domain-invariant shared features across networks. SOGA (Mao et al., 2024) first time uses discriminability by encouraging the structural consistencies between target nodes in the same class for the SFDA in the graph. GraphAE (Guo et al., 2022) focuses on how shifts in node degree distribution affect node embeddings by minimizing the discrepancy between router embedding to eliminate structural shifts. SpecReg (You et al., 2022) used the optimal transport-based GDA bound for graph data and discovered that revising the GNNs' Lipschitz constant can be achieved by spectral smoothness and maximum frequency response. JHGDA (Shi et al., 2023) studies the shifts in hierarchical graph structures, which are inherent properties of graphs by aggregating domain discrepancy from all hierarchy levels to derive a comprehensive discrepancy measurement. ALEX (Yuan et al., 2023) first creates a label shift enhanced augmented graph view using a low-rank adjacency matrix obtained through singular value decomposition by driving contrasting loss. SGDA (Qiao et al., 2023) enhances original source graphs by integrating trainable perturbations (adaptive shift parameters) into embeddings by conducting adversarial learning to simultaneously train both the graph encoder and perturbations, to minimize marginal shifts.

## 3 THEORETICAL ANALYSIS

In this subsection, we provide a discussion on the PAC-Bayesian analysis with the graph domain adaptation.

**Notations.** An undirected graph $G = \{\mathcal{V}, \mathcal{E}, A, X, Y\}$ consists of a set of nodes $\mathcal{V}$ and edges $\mathcal{E}$, along with an adjacency matrix $A$, a feature matrix $X$, and a label matrix $Y$. The adjacency matrix $A \in \mathbb{R}^{N \times N}$ encodes the connections between $N$ nodes, where $A_{ij} = 1$ indicates an edge between nodes $i$ and $j$, and $A_{ij} = 0$ means the nodes are not connected. The feature matrix $X \in \mathbb{R}^{N \times d}$ represents the node features, with each node described by a $d$-dimensional feature vector. Finally, $Y \in \mathbb{R}^{N \times C}$ contains the labels for the $N$ nodes, where each node is classified into one of $C$ classes.

In this work, we explore the task of node classification in a unsupervised setting, where both the node feature matrix $X$ and the graph structure $A$ are given before learning. We assume that all key aspects of our analysis are conditioned on the fixed graph structure $A$ and feature matrix $X$, while the uncertainty arises from the node labels $Y$. Specifically, we assume that the label $y_i$ for each node $i \in \mathcal{V}$ is drawn from a latent conditional distribution $\Pr(y_i \mid Z_i)$, where $Z = f(X, G)$, with $f$ being an aggregation function that combines features from the local neighborhood of each node within the graph. Additionally, we assume that the labels for different nodes are independent of each other, given their respective aggregated feature representations $Z_i$. With a partially labeled node set $V_0 \subseteq \mathcal{V}$, our objective in the node classification problem is to learn a model $h : \mathbb{R}^{N \times d} \times G_N \rightarrow \mathbb{R}^{N \times C}$ from a family of classifiers $\mathcal{H}$ that can predict the labels for the remaining unlabeled nodes. For a given classifier $h$, the predicted label $\hat{Y}_i$ for node $i$ is determined by: $\hat{Y}_i = \arg\max_{k \in \{1,...,C\}} h_i(X, G)[k]$,

where $h_i(X, G)$ is the output corresponding to node $i$ and $h_i(X, G)[k]$ represents the score for the $k$-th class for node $i$.

**Margin loss on each domain.** Now we can define the empirical and expected margin loss of a classifier $h \in \mathcal{H}$ on source graph $G^S = \{\mathcal{V}^S, \mathcal{E}^S, A^S, X^S, Y^S\}$ and target graph $G^T = \{\mathcal{V}^T, \mathcal{E}^T, A^T, X^T\}$. Given $Y^S$, the empirical margin loss of $h$ on $G^S$ for a margin $\gamma \geq 0$ is defined as

$$\widehat{\mathcal{L}}_S^\gamma(h) := \frac{1}{N_S} \sum_{i \in \mathcal{V}^S} \mathbb{1}\left[h_i(X^S, G^S)[Y_i] \leq \gamma + \max_{c \neq Y_i} h_i(X^S, G^S)[c]\right] \tag{1}$$

where $\mathbb{1}[\cdot]$ is the indicator function, $c$ represents node labeling . The expected margin loss is then defined as

$$\mathcal{L}_S^\gamma(h) := \mathbb{E}_{Y_i \sim \mathrm{Pr}(Y|Z_i), i \in \mathcal{V}^S} \widehat{\mathcal{L}}_S^\gamma(h) \tag{2}$$

**Definition 1** (Expected Loss Discrepancy). *Given a distribution $P$ over a function family $\mathcal{H}$, for any $\lambda > 0$ and $\gamma \geq 0$, for any $G^S$ and $G^T$, define the expected loss discrepancy between $\mathcal{V}^S$ and $\mathcal{V}^T$ as $D_{S,T}^\gamma(P; \lambda) := \ln \mathbb{E}_{h \sim P} e^{\lambda\left(\mathcal{L}_T^{\gamma/2}(h) - \mathcal{L}_S^\gamma(h)\right)}$, where $\mathcal{L}_T^{\gamma/2}(h)$ and $\mathcal{L}_S^\gamma(h)$ follow the definition of Eq. (2).*

Intuitively, $D_{S,T}^\gamma(P; \lambda)$ captures the difference of the expected loss between $\mathcal{V}^S$ and $\mathcal{V}^T$ in an average sense (over $P$).

**Theorem 1** (Domain Adaptation Bound for Deterministic Classifiers). *Let $\mathcal{H}$ be a family of classification functions. For any classifier $h$ in $\mathcal{H}$, and for any parameters $\lambda > 0$ and $\gamma \geq 0$, consider any prior distribution $P$ over $\mathcal{H}$ that is independent of the training data $\mathcal{V}^S$. With a probability of at least $1 - \delta$ over the sample $Y^S$, for any distribution $Q$ on $\mathcal{H}$ such that $\mathrm{Pr}_{\tilde{h} \sim Q}\left[\max_{i \in \mathcal{V}^S \cup \mathcal{V}^T} \|h_i(X, G) - \tilde{h}_i(X, G)\|_\infty < \frac{\gamma}{8}\right] > \frac{1}{2}$, the following inequality holds:*

$$\mathcal{L}_T^0(\tilde{h}) \leq \widehat{\mathcal{L}}_S^\gamma(\tilde{h}) + \frac{1}{\lambda}\left[2(D_{\mathrm{KL}}(Q\|P) + 1) + \ln\frac{1}{\delta} + \frac{\lambda^2}{4N_S} + D_{S,T}^{\gamma/2}(P; \lambda)\right]. \tag{3}$$

We follow the characterization from (Ma et al., 2021). In the generalization bound, the KL-divergence $D_{\mathrm{KL}}(Q\|P)$ is usually considered as a measurement of the model complexity. The terms $\ln(1/\delta)$ and $\frac{\lambda^2}{4N_S}$ are commonly seen in PAC-Bayesian analysis for IID supervised settings. The expected loss discrepancy $D_{S,T}^{\gamma/2}(P; \lambda)$ between the source nodes $\mathcal{V}^S$ and the targeted nodes $\mathcal{V}^T$ is essential to our analysis. To derive the generalization guarantee, we need to upper-bound the expected loss discrepancy $D_{S,T}^\gamma(P; \lambda)$.

**Proposition 1** (Bound for $D_{S,T}^\gamma(P; \lambda)$). *For any $\gamma \geq 0$, and under the assumption that the prior distribution $P$ over the classification function family $\mathcal{H}$ is defined, we establish a bound for the domain discrepancy measure $D_{S,T}^{\gamma/2}(P; \lambda)$. Specifically, we have the following inequality:*

$$D_{S,T}^{\gamma/2}(P; \lambda) \leq O\left(\sum_{i \in V^S} \sum_{j \in V^T} \|(A^S X^S)_i - (A^T X^T)_j\|_2^2 + \sum_{i \in V^S} \sum_{j \in V^T} \|X_i^S - X_j^T\|_2^2\right). \tag{4}$$

From Proposition 1, the topological divergence $\|(A^S X^S)_i - (A^T X^T)_j\|_2^2$ and the attribute divergence $\|X_i^S - X_j^T\|_2^2$ constitute the upper bound of $D_{S,T}^\gamma(P; \lambda)$ and further bound graph domain discrepancy. It also reveals both attribute and topology divergence of intrinsic graph property influence on GDA generalization upper-bound. We introduce attribute and topology alignment loss by minimizing attribute divergence by utilizing the attribute graph and minimizing topology divergence by utilizing the original graph. Graph $G^S$ and $G^T$ with topology information $A^S$ and $A^T$ through the feature extraction module can represent $(A^S X^S)_i$ and $(A^S X^S)_j$.

## 4 THE PROPOSED METHODOLOGY

In this section, we propose a novel GDA method with attribute-driven alignment (GAA), which first minimizes graph attribute divergence. The overall framework of GAA is shown in Figure 2. The main components of the proposed method include the specific attribute convolution module and the attribute-driven alignment module. We will detail the proposed GAA in the following subsections.

### 4.1 SPECIFIC ATTRIBUTE CONVOLUTION MODULE

Inspired by Proposition 1, we design an attribute-driven GDA model by using topology graph and feature graph. Our model mainly contains attribute-driven alignment that directly minimize discrimination in attribute and topology between source and target graph.

**Feature Graph** Merely using node attribute information through $X$ is unstable (Fang et al., 2022; Mao et al., 2023; Li et al., 2024; Xie et al., 2024b). A natural idea would be to utilize graph node attribute by fully making use of the information through feature space propagation (Wang et al., 2020; Xie et al., 2024a; Kang et al., 2024; Li et al., 2025). Therefore we introduce feature graph into our work.

To represent the structure of nodes in the feature space, we build a $k$NN graph $\hat{G}$ based on the feature matrix $X$. To be precise, a node similarity matrix $SM$ is computed using the cosine similarity formula:

$$SM_{ij} = \frac{X_i \cdot X_j}{|X_i| \cdot |X_j|} \tag{5}$$

where $SM_{ij}$ is the similarity between node feature $X_i$ and node feature $X_j$. We derivate feature graph $\hat{G} = \left\{ \mathcal{V}, \hat{\mathcal{E}}, \hat{A}, X, Y \right\}$, which shares the same $X$ with $G$, but has a different adjacency matrix. Therefore, topology graph and feature graph refer to $G$ and $\hat{G}$ respectively. Then for each node we choose the top k nearest neighbors and establish edges. In this way, we construct a feature graph in attribute view for the source graph $\hat{G}^S = \left\{ \mathcal{V}^S, \hat{\mathcal{E}}^S, \hat{A}^S, X^S, Y^S \right\}$ and target graph $\hat{G}^T = \left\{ \mathcal{V}^T, \hat{\mathcal{E}}^T, \hat{A}^T, X^T \right\}$.

**Feature Extraction Module** To extract meaningful features from graphs, we adopt GCN that is comprised of multiple graph convolutional layers. With the input graph $G$, the $(l+1)$-th layer's output $H^{(l+1)}$ can be represented as:

$$H^{(l+1)} = ReLU(D^{-\frac{1}{2}} A D^{-\frac{1}{2}} H^{(l)} W^{(l)}) \tag{6}$$

where $ReLU$ is the Relu activation function ($ReLU(\cdot) = max(0, \cdot)$), $D$ is the degree matrix of $A$, $W^{(l)}$ is a layer-specific trainable weight matrix, $H^{(l)}$ is the activation matrix in the $l$-th layer and $H^{(0)} = X$. In our study we use two GCNs to exploit the information in topology and feature space. For source graph, output node embedding is donated by $Z^S$ generated from $G^S$ and $Z_f^S$ generated from $\hat{G}^S$. Similarly, for the target graph, the output node embedding is donated by $Z^T$ generated from $G^T$ and $Z_f^T$ generated from $\hat{G}^T$.

### 4.2 SOURCE CLASSIFIER LOSS

The source classifier loss $\mathcal{L}_S \left( f_S \left( Z^S \right), Y^S \right)$ is to minimize the cross-entropy for the labeled data node in the source domain:

$$\mathcal{L}_S \left( f_S \left( Z^S \right), Y^S \right) = -\frac{1}{N_S} \sum_{i=1}^{N_S} y_i^S \log \left( \hat{y}_i^S \right) \tag{7}$$

where $y_i^S$ denotes the label of the $i$-th node in the source domain and $\hat{y}_i^S$ are the classification prediction for the $i$-th source graph labeled node $v_i^S \in \mathcal{V}^S$.

### 4.3 ATTRIBUTE-DRIVEN ALIGNMENT

To make the attribute view fully learnable, we design the attention attribute module to dynamically utilize the important attribute. Specifically, we design learnable domain adaptive models for alignment embeddings in topology and attribute views.

**Attention-based Attribute** To guide the network to take more attention to the important node attributes and make attributes learnable, we design attention-based embedding models. Specifically, we map the node attributes into three different latent spaces. By given an example in source graph attribute embedding: $Q = W_q Z_f^{S\top}$, $K = W_k Z_f^{S\top}$, $M = W_v Z_f^{S\top}$, where $W_q \in \mathbb{R}^{d \times d}, W_k \in \mathbb{R}^{d \times d}, W_v \in \mathbb{R}^{d \times d}$ are the learnable parameter matrices. And $Q \in \mathbb{R}^{d \times N}, K \in \mathbb{R}^{d \times N}$ and $M \in \mathbb{R}^{d \times N}$ denotes the query matrix, key matrix and value matrix, respectively.

The attention-based attribute matrix $att_f^S$ can be calculated by:

$$att_f^S = softmax\left(\frac{K^\top Q}{\sqrt{d}}\right) M^\top \tag{8}$$

Likewise, we can obtain a similar objective of each learnable graph embedding $att^S$, $att_f^T$ and $att^T$.

**Cross-view Similarity Matrix Refinement**

Subsequently, the cross-view similarity matrix $S^S$ represents the similarity between the source attribute and topology graph. $S^T$ represents the similarity between the target attribute and topology graph. $S^S$ as formulated:

$$S^S = \frac{Z_f^S \cdot \left(Z^S\right)^\top}{||Z_f^S||_2 \cdot ||Z^S||_2} \tag{9}$$

Likewise, we can obtain a similarity matrix of the target graph by:

$$S^T = \frac{Z_f^T \cdot \left(Z^T\right)^\top}{||Z_f^T||_2 \cdot ||Z^T||_2} \tag{10}$$

where $S^S$ and $S^T$ is the cross-view similarity matrix, and $\langle \cdot \rangle$ is the function to calculate similarity. Here, we adopt cosine similarity Qian et al. (2024). The proposed similarity matrix $S^S$ and $S^T$ measures the similarity between samples by comprehensively considering attribute and structure information. The connected relationships between different nodes could be reflected by $S^S$ and $S^T$. Therefore, we utilize $S^T$ and $S^S$ to refine the structure in augmented view with Hadamard product, $att_f^S$ can be formulated as:

$$att^S = att^S \odot S^S \tag{11}$$

Similarly we can get $att_f^S$ by $att_f^S \odot S^S$, $att_f^T$ by $att_f^T \odot S^T$, $att^T$ by $att^T \odot S^T$, which respectively represent source graph and target graph in both topology view and attribute view embedding.

**Attribute-Driven Domain Adaptive**

The proposed framework follows the transfer learning paradigm, where the model minimizes the divergence of the two views. In detail, GAA jointly optimizes two views of GDA alignment. To be specific, $\mathcal{L}_A$ is the Mean Squared Error (MSE) loss between the source graph $att^S$ and $att_f^S$ and the target graph $att^T$ and $att_f^T$, which can be formulated as:

$$\mathcal{L}_A = -\left(||att^S - att^T||_2^2 + ||att_f^S - att_f^T||_2^2\right) \tag{12}$$

We adapt the domain in two views, domain classifier loss in the topology view is $||att_f^S - att_f^T||_2^2$ enforces that the attribute graph node representation after the node feature extraction and similarity matrix refinement from source and target graph $G_f^S$ and $G_f^T$. Similarly, we get $||att^S - att^T||_2^2$

from $G^S$ and $G^T$. And $||att^S - att^T||_2^2$ corresponds to the first item of Proposition 1, which is $||(A^S X^S)_i - (A^T X^T)_j||_2^2$ means minimizing structural distribution shift. In attribute view is $||att_f^S - att_f^T||_2^2$ trying to discriminate corresponds to the second term $||X_i^S - X_j^T||_2^2$ of Proposition 1, which means minimizing attribute distribution shift.

## 4.4 TARGET NODE CLASSIFICATION

We use Gradient Reversal Layer (GRL) (Ganin et al., 2016) for adversarial training. Mathematically, we define the GRL as $Q_\lambda(x) = x$ with a reversal gradient $\frac{\partial Q_\lambda(x)}{\partial x} = -\lambda I$. Learning a GRL is adversarial in such a way that: on the one side, the reversal gradient enforces $f_S(Z^S)$ to be maximized; on the other side, $\theta_D$ is optimized by minimizing the cross-entropy domain classifier loss:

$$\mathcal{L}_D = -\frac{1}{N_S + N_T} \sum_{i=1}^{N_S+N_T} m_i \log(\hat{m}_i) + (1 - m_i) \log(1 - \hat{m}_i) \tag{13}$$

where $m_i \in \{0, 1\}$ denotes the groundtruth, and $\hat{m}_i$ denotes the domain prediction for the $i$-th node in the source domain and target domain, respectively. To utilize the data in the target domain, we use entropy loss for the target classifier $f_T$:

$$\mathcal{L}_T\left(f_T\left(Z^T\right)\right) = -\frac{1}{N_T} \sum_{i=1}^{N_T} \hat{y}_i^T \log\left(\hat{y}_i^T\right) \tag{14}$$

where $\hat{y}_i^T$ are the classification prediction for the $i$-th node in the target graph $v_i^T$. Finally, by combining $\mathcal{L}_A$, $\mathcal{L}_S$, $\mathcal{L}_D$ and $\mathcal{L}_T$, the overall loss function of our model can be represented as:

$$\mathcal{L} = \mathcal{L}_A + \alpha \mathcal{L}_S + \beta \mathcal{L}_D + \tau \mathcal{L}_T \tag{15}$$

where $\alpha$, $\beta$ and $\tau$ are trade-off hyper-parameters. The parameters of the whole framework are updated via backpropagation.

# 5 EXPERIMENT

## 5.1 DATASETS

To prove the superiority of our work on domain adaptation node classification tasks, we evaluate it on four types of datasets, including Airport dataset (Ribeiro et al., 2017), Citation dataset (Wu et al., 2020), Social dataset (Liu et al., 2024a) and Blog dataset (Li et al., 2015). The airport dataset involves three countries' airport traffic networks: USA (U), Brazil (B), and Europe (E), in which the node indicates the airport and the edge indicates the routes between two airports. The citation dataset includes three different citation networks: DBLPv8 (D) , ACMv9 (A), and Citationv2 (C), in which the node indicates the article and the edge indicates the citation relation between two articles. As for social networks, we choose Twitch gamer networks and Blog Network, which are collected from Germany(DE) and England(EN). Two disjoint Blog social networks, Blog1 (B1) and Blog2 (B2), which are extracted from the BlogCatalog dataset. extracted from the BlogCatalog dataset. Because these four groups of dataset ingredients are generated from different data sources, their distributions are naturally diverse. For a comprehensive overview of these datasets, please refer to Tab 5.

| Types | Datasets | #Node | #Edge | #Label |
|---|---|---|---|---|
| Airport | USA | 1,190 | 13,599 | 4 |
| | Brazil | 131 | 1,038 | |
| | Europe | 399 | 5,995 | |
| Citation | ACMv9 | 9,360 | 15,556 | 5 |
| | Citationv1 | 8,935 | 15,098 | |
| | DBLPv7 | 5,484 | 8,117 | |
| Social | Blog1 | 2,300 | 33,471 | 6 |
| | Blog2 | 2,896 | 53,836 | |
| Social | Germany | 9,498 | 153,138 | 2 |
| | England | 7,126 | 35,324 | |
| MAG | US | 132,558 | 697,450 | 20 |
| | CN | 101,952 | 285,561 | |
| | DE | 43,032 | 126,683 | |
| | JP | 37,498 | 90,944 | |
| | RU | 32,833 | 67,994 | |
| | FR | 29,262 | 78,222 | |

Table 1: Dataset Statistics.

| Methods | U → B | U → E | B → U | B → E | E → U | E → B | DE → EN | EN → DE |
|---|---|---|---|---|---|---|---|---|
| GCN | 0.366 | 0.371 | 0.491 | 0.452 | 0.439 | 0.298 | 0.673 | 0.634 |
| $k$NN-GCN | 0.436 | 0.437 | 0.461 | 0.478 | 0.459 | 0.464 | 0.661 | 0.623 |
| DANN | 0.501 | 0.386 | 0.402 | 0.350 | 0.436 | 0.538 | 0.512 | 0.528 |
| DANE | 0.531 | 0.472 | 0.491 | 0.489 | 0.461 | 0.520 | 0.642 | 0.644 |
| UDAGCN | 0.607 | 0.488 | 0.497 | 0.510 | 0.434 | 0.477 | 0.724 | 0.660 |
| ASN | 0.519 | 0.469 | 0.498 | 0.494 | 0.466 | 0.595 | 0.550 | 0.679 |
| EGI | 0.523 | 0.451 | 0.417 | 0.454 | 0.452 | 0.588 | 0.681 | 0.589 |
| GRADE-N | 0.550 | 0.457 | 0.497 | 0.506 | 0.463 | 0.588 | 0.749 | 0.661 |
| JHGDA | 0.695 | 0.519 | 0.511 | 0.569 | 0.522 | **0.740** | 0.766 | 0.737 |
| SpecReg | 0.481 | 0.487 | 0.513 | 0.546 | 0.436 | 0.527 | 0.756 | 0.678 |
| GIFI | 0.636 | 0.521 | 0.493 | 0.535 | 0.501 | 0.623 | 0.719 | 0.705 |
| PA | 0.679 | 0.557 | 0.528 | 0.562 | 0.547 | 0.529 | 0.677 | 0.760 |
| **GAA** | **0.704** | **0.563** | **0.542** | **0.573** | **0.546** | 0.691 | **0.779** | **0.751** |

Table 2: Cross-network node classification on the Airport network.

| Methods | A → D | D → A | A → C | C → A | C → D | D → C | B1 → B2 | B2 → B1 |
|---|---|---|---|---|---|---|---|---|
| GCN | 0.632 | 0.578 | 0.675 | 0.635 | 0.666 | 0.654 | 0.408 | 0.451 |
| $k$NN-GCN | 0.636 | 0.587 | 0.672 | 0.648 | 0.668 | 0.426 | 0.531 | 0.579 |
| DANN | 0.488 | 0.436 | 0.520 | 0.518 | 0.518 | 0.465 | 0.409 | 0.419 |
| DANE | 0.664 | 0.619 | 0.642 | 0.653 | 0.661 | 0.709 | 0.464 | 0.423 4 |
| UDAGCN | 0.684 | 0.623 | 0.728 | 0.663 | 0.712 | 0.645 | 0.471 | 0.468 |
| ASN | 0.729 | 0.723 | 0.752 | 0.678 | 0.752 | 0.754 | 0.732 | 0.524 |
| EGI | 0.647 | 0.557 | 0.676 | 0.598 | 0.662 | 0.652 | 0.494 | 0.516 |
| GRADE-N | 0.701 | 0.660 | 0.736 | 0.687 | 0.722 | 0.687 | 0.567 | 0.541 |
| JHGDA | 0.755 | 0.737 | 0.814 | 0.756 | 0.762 | 0.794 | 0.619 | 0.643 |
| SpecReg | 0.762 | 0.654 | 0.753 | 0.680 | 0.768 | 0.727 | 0.661 | 0.631 |
| GIFI | 0.751 | 0.737 | 0.793 | 0.755 | 0.739 | 0.751 | 0.653 | 0.642 |
| PA | 0.752 | 0.751 | 0.804 | 0.768 | 0.755 | 0.780 | 0.662 | 0.654 |
| **GAA** | **0.789** | **0.754** | **0.824** | **0.782** | **0.771** | **0.798** | **0.681** | **0.679** |

Table 3: Cross-network node classification on the Citation, Blog and Social network.

| Methods | US → CN | US → DE | US → JP | US → RU | US → FR | CN → US | CN → DE | CN → JP | CN → RU | CN → FR |
|---|---|---|---|---|---|---|---|---|---|---|
| GCN | 0.042 | 0.168 | 0.219 | 0.147 | 0.182 | 0.193 | 0.064 | 0.160 | 0.069 | 0.067 |
| $k$NN-GCN | 0.092 | 0.189 | 0.269 | 0.186 | 0.213 | 0.210 | 0.133 | 0.201 | 0.105 | 0.102 |
| DANN | 0.242 | 0.263 | 0.379 | 0.218 | 0.207 | 0.302 | 0.134 | 0.214 | 0.119 | 0.107 |
| DANE | 0.272 | 0.250 | 0.280 | 0.210 | 0.186 | 0.279 | 0.108 | 0.228 | 0.170 | 0.184 |
| UDAGCN | OOM | OOM | OOM | OOM | OOM | OOM | OOM | OOM | OOM | OOM |
| ASN | 0.290 | 0.272 | 0.291 | 0.222 | 0.199 | 0.268 | 0.121 | 0.207 | 0.189 | 0.190 |
| EGI | OOM | OOM | OOM | OOM | OOM | OOM | OOM | OOM | OOM | OOM |
| GRADE-N | 0.304 | 0.299 | 0.306 | 0.240 | 0.217 | 0.258 | 0.137 | 0.210 | 0.178 | 0.199 |
| JHGDA | OOM | OOM | OOM | OOM | OOM | OOM | OOM | OOM | OOM | OOM |
| SpecReg | 0.237 | 0.267 | 0.377 | 0.228 | 0.218 | 0.317 | 0.134 | 0.199 | 0.109 | 116 |
| PA | 0.400 | 0.389 | 0.474 | 0.371 | 0.252 | 0.452 | 0.262 | 0.383 | 0.333 | 0.242 |
| **GAA** | **0.410** | **0.401** | **0.492** | **0.372** | **0.2881** | **0.453** | **0.302** | **0.400** | **0.351** | **0.293** |

Table 4: Cross-network node classification on MAG datasets.

## 5.2 BASELINES

We choose some representative methods to compare. **GCN** (Kipf & Welling, 2016) further solves the efficiency problem by introducing first-order approximation of ChebNet. $k$**NN-GCN** (Wang et al., 2020) use the sparse $k$-nearest neighbor graph calculated from feature matrix as the input graph of GCN and name it $k$NN-GCN. **DANN** (Ganin et al., 2016) use a 2-layer perceptron to provide features and a gradient reverse layer (GRL) to learn node embeddings for domain classification **DANE** (Zhang et al., 2019) shared distributions embedded space on different networks and further aligned them through adversarial learning regularization. **UDAGCN** (Wu et al., 2020) is a dual graph convolutional network component learning framework for unsupervised GDA, which captures

knowledge from local and global levels to adapt it by adversarial training. **ASN** (Zhang et al., 2021) use the domain-specific features in the network to extract the domain-invariant shared features across networks. **EGI** (Zhu et al., 2021) through Ego-Graph Information maximization to analyze structure-relevant transferability regarding the difference between source-target graph. **GRADE-N** (Wu et al., 2023) propose a graph subtree discrepancy to measure the graph distribution shift between source and target graphs. **JHGDA** (Shi et al., 2023) explore information from different levels of network hierarchy by hierarchical pooling model. **SpecReg** (You et al., 2022) achieve improving performance regularization inspired by cross-pollinating between the optimal transport DA and graph filter theories . **GIFI** (Qiao et al., 2024) uses a parameterized graph reduction module and variational information bottleneck to filter out irrelevant information. **PA** (Liu et al., 2024b) mitigates distribution shifts in graph data by recalibrating edge influences to handle structure shifts and adjusting classification losses to tackle label shifts.

### 5.3 EXPERIMENTAL SETUP

The experiments are implemented in the PyTorch platform using an Intel(R) Xeon(R) Silver 4210R CPU @ 2.40GHz, and GeForce RTX A5000 24G GPU. Technically, two layers GCN is built and we train our model by utilizing the Adam (Kingma & Ba, 2015) optimizer with learning rate ranging from 0.0001 to 0.0005. In order to prevent over-fitting, we set the dropout rate to 0.5. In addition, we set weight decay $\in \{1e-4, \cdots, 5e-3\}$ and $k \in \{1, \cdots, 10\}$ for $k$NN graph. For fairness, we use the same parameter settings for all the cross-domain node classification methods in our experiment, except for some special cases. For GCN, UDA-GCN, and JHGDA the GCNs of both the source and target networks contain two hidden layers ($L = 2$) with structure as $128 - 16$. The dropout rate for each GCN layer is set to $0.3$. We repeatedly train and test our model for five times with the same partition of dataset and then report the average of ACC.

### 5.4 CROSS-NETWORK NODE CLASSIFICATION RESULTS

The results of experiments are summarized in Table 2 and 4, where the best performance is highlighted in boldface. Some results are directly taken from (Shi et al., 2023; Pang et al., 2023). We have the following findings: It can be seen that our proposed method boosts the performance of SOTA methods across most evaluation metrics on four group datasets with 16 tasks, which proves its effectiveness. Particularly, compared with other optimal performances in all datasets, GAA achieves a maximum average improvement of $1.80\%$ for ACC. This illustrates that our proposed model can effectively utilize node attribute information. Our GAA achieves much better performances than SpecReg and JHGDA on all of the metrics in a dataset of Airport and most of the metrics in a dataset of Citation. This can be explained by our method's use of attribute and topology structure. In most cases, GAA produces better performance than GRADE- N (Wu et al., 2023) and JHGDA (Shi et al., 2023), which were published in 2023. This verifies the advantage of our approach. On most occasions, the feature graph produces a better result than the original graph. For example, in airport data, $k$NN-GCN performance averages better than $5.30\%$ to GCN, and in citation datasets, performance averages better than $0.60\%$ to GCN. Our findings affirm that the observed discrepancy in node attributes surpasses that of the topological misalignment, thus suggesting that the alignment of node attributes holds potential for yielding more substantial enhancements.

### 5.5 ABLATION STUDY

To validate the effectiveness of different components in our model, we compare GAA with its three variants on Citation and Airport datasets.

- **GAA$_1$**: GAA without cross-view similarity matrix Refinement to show the importance of comprehensive attribute and structure information.
- **GAA$_2$**: GAA without $\mathcal{L}_A$ to show the impact of attribute benefit alignment.
- **GAA$_3$**: GAA without $\mathcal{L}_A$ and remove channel feature graph and only utilize $\mathcal{L}_D$ to show the effect of attribute(feature) graph impact.

According to Figure3, we can draw the following conclusions: (1) The results of GAA are consistently better than all variants, indicating the rationality of our model. (2) Both topology and feature

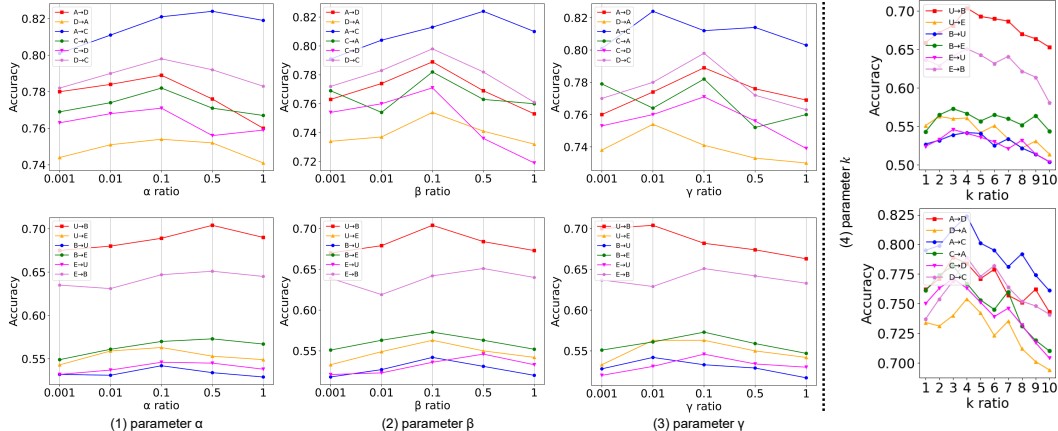

Figure 4: The influence of parameters $\alpha$, $\beta$, $\tau$ and $k$ on Citation and Airport dataset.

information are crucial to domain adaptation. (3) The cross-view similarity matrix can improve performance by enhancing and integrating feature information, benefiting the synergistic refinement of both attribute and topology.

## 5.6 PARAMETER ANALYSIS

In this section, we analyze the sensitivity of the parameters of our method on the Airport dataset and Citation dataset. As shown in Figure.5 in Subfigure (4), the accuracy usually peaks at $2-3$ with $k$. This is reasonable since increasing $k$ means more high-order proximity information is incorporated. On the other hand, extremely large $k$ could also introduce noise that will deteriorate the performance. Furthermore, each dataset may

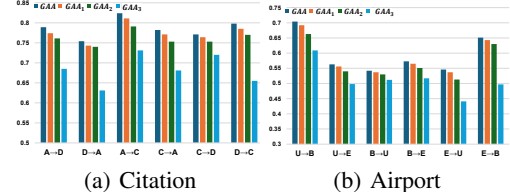

(a) Citation    (b) Airport

Figure 3: The classification accuracy of GAA and its variants on citation datasets and airport dataset.

require a unique optimal value for $k$, determined by its inherent properties. For more information, please refer to the Appendix D. From Figure.5 Subfigure (1) (2) (3), we can see GAA has competitive performance on a large range of values, which suggests the stability of our method.

## 6 CONCLUSION

In this paper, we propose GAA framework to solve the GDA problem in cross-network node classification tasks. We reveal the importance of both attributes and topology in GDA through both empirical and theoretical analysis, which minimizes attribute and topology distribution shifts based on intrinsic graph properties. Comprehensive experiments verify the superiority of our approach. In the future, we may strive to design new frameworks for other cross-network learning tasks, including link-level and graph-level. We will also deep into graph domain adaptation theory for developing more powerful models.

## 7 ACKNOWLEDGMENTS

This work is supported by the Natural Sciences and Engineering Research Council of Canada (NSERC) Discovery Grants program. This work is also supported by the Vector Scholarship in Artificial Intelligence, provided through the Vector Institute.

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

## A  PROOF OF PROPOSITION 1

To facilitate the analysis, we adopt the following data assumption:

**Definition 1.** *The generated nodes consist of two disjoint sets, denoted as $c_0$ and $c_1$. Each node feature $x$ is sampled from $N(\mu_i, \sigma_i)$ for $i \in \{0, 1\}$.*

*Each set $c_i$ corresponds to the source graph and target graph compositions, respectively: $c_i^{(S)}$ and $c_i^{(T)}$. The class distribution is balanced, such that $\mathbb{P}(\mathbb{Y} = c_0) = \mathbb{P}(\mathbb{Y} = c_1)$.*

**Theorem 2.** *For nodes $s \in V_S$ and $t \in V_T$ with aggregated features $\mathbf{f} = GNN(x)$, the following inequality holds:*

$$||\mathbb{P}(y_u = c_0|\mathbf{f}_u) - \mathbb{P}(y_v = c_0|\mathbf{f}_v)|| \leq O(||\mathbf{f}_u - \mathbf{f}_v|| + ||\mathbf{f}_v - \boldsymbol{\mu}_1^{(S)}|| + ||\mathbf{f}_v - \boldsymbol{\mu}_1^{(T)}||). \quad (16)$$

*Proof.* The conditional probability of class $c_0$ given the aggregated feature $f$ can be expressed using Bayes' theorem:

$$\mathbb{P}(\mathbf{y}_s = c_0|\mathbf{f}_s) = \frac{\mathbb{P}(\mathbf{f}_s|\mathbf{y}_s = c_0)\mathbb{P}(\mathbf{y}_s = c_0)}{\mathbb{P}(\mathbf{f}_s|\mathbf{y}_s = c_0)\mathbb{P}(\mathbf{y}_s = c_0) + \mathbb{P}(\mathbf{f}_s|\mathbf{y}_s = c_1)\mathbb{P}(\mathbf{y}_s = c_1)}. \quad (17)$$

Under the assumption $\mathbb{P}(\mathbf{y} = c_0) = \mathbb{P}(\mathbf{y} = c_1)$, we simplify this to:

$$\mathbb{P}(\mathbf{y}_s = c_0|\mathbf{f}_s) = \frac{\mathbb{P}(\mathbf{f}_s|\mathbf{y}_s = c_0)}{\mathbb{P}(\mathbf{f}_s|\mathbf{y}_s = c_0) + \mathbb{P}(\mathbf{f}_s|\mathbf{y}_s = c_1)}. \quad (18)$$

Substituting in the expressions for the Gaussian distributions:

$$\mathbb{P}(\mathbf{y}_s = c_0|\mathbf{f}_s) = \frac{\exp\left(-\frac{(\mathbf{f}_u - \boldsymbol{\mu}_0^{(S)})^2}{\sigma^2}\right)}{\exp\left(-\frac{(\mathbf{f}_u - \boldsymbol{\mu}_0^{(S)})^2}{\sigma^2}\right) + \exp\left(-\frac{(\mathbf{f}_u - \boldsymbol{\mu}_1^{(S)})^2}{\sigma^2}\right)}. \quad (19)$$

Thus, we have:

$$\begin{aligned}
&||\mathbb{P}(\mathbf{y}_u = c_0|\mathbf{f}_u) - \mathbb{P}(\mathbf{y}_v = c_0|\mathbf{f}_v)|| \\
&= ||\frac{\mathbb{P}(\mathbf{f}_s|\mathbf{y}_s = c_0)}{\mathbb{P}(\mathbf{f}_s|\mathbf{y}_s = c_0) + \mathbb{P}(\mathbf{f}_s|\mathbf{y}_s = c_1)} - \frac{\mathbb{P}(\mathbf{f}_v|\mathbf{y}_v = c_0)}{\mathbb{P}(\mathbf{f}_v|\mathbf{y}_v = c_0) + \mathbb{P}(\mathbf{f}_v|\mathbf{y}_v = c_1)}|| \\
&= \frac{||\mathbb{P}(\mathbf{f}_s|\mathbf{y}_s = c_0)\mathbb{P}(\mathbf{f}_v|\mathbf{y}_v = c_1) - \mathbb{P}(\mathbf{f}_v|\mathbf{y}_v = c_0)\mathbb{P}(\mathbf{f}_s|\mathbf{y}_s = c_1)||}{[\mathbb{P}(\mathbf{f}_s|\mathbf{y}_s = c_0) + \mathbb{P}(\mathbf{f}_s|\mathbf{y}_s = c_1)][\mathbb{P}(\mathbf{f}_v|\mathbf{y}_v = c_0) + \mathbb{P}(\mathbf{f}_v|\mathbf{y}_v = c_1)]}.
\end{aligned} \quad (20)$$

Noting that the denominator is bounded, we substitute the probabilities of the Gaussian distributions into the expression:

$$\begin{aligned}
&||\mathbb{P}(\mathbf{y}_u = c_0|\mathbf{f}_u) - \mathbb{P}(\mathbf{y}_v = c_0|\mathbf{f}_v)|| \\
&= \frac{||\exp\left(-\frac{(\mathbf{f}_u - \boldsymbol{\mu}_0^{(S)})^2}{\sigma^2}\right)\exp\left(-\frac{(\mathbf{f}_v - \boldsymbol{\mu}_1^{(T)})^2}{\sigma^2}\right) - \exp\left(-\frac{(\mathbf{f}_u - \boldsymbol{\mu}_1^{(S)})^2}{\sigma^2}\right)\exp\left(-\frac{(\mathbf{f}_v - \boldsymbol{\mu}_0^{(T)})^2}{\sigma^2}\right)||}{\exp(-A)}.
\end{aligned} \quad (21)$$

This leads us to:

$$||\mathbb{P}(\mathbf{y}_u = c_0|\mathbf{f}_u) - \mathbb{P}(\mathbf{y}_v = c_0|\mathbf{f}_v)|| \leq \frac{1}{\sigma^2}||(\boldsymbol{\mu}_0^{(T)} - \boldsymbol{\mu}_0^{(S)})(2\mathbf{f}_u - \boldsymbol{\mu}_0^{(S)} - \boldsymbol{\mu}_0^{(T)}) - (\boldsymbol{\mu}_1^{(T)} - \boldsymbol{\mu}_1^{(S)})(2\mathbf{f}_v - \boldsymbol{\mu}_1^{(S)} - \boldsymbol{\mu}_1^{(T)})||. \quad (22)$$

This simplifies to:

$$||\mathbb{P}(\mathbf{y}_u = c_0|\mathbf{f}_u) - \mathbb{P}(\mathbf{y}_v = c_0|\mathbf{f}_v)|| \le O(||\mathbf{f}_u - \mathbf{f}_v|| + ||2\mathbf{f}_v - \boldsymbol{\mu}_1^{(S)} - \boldsymbol{\mu}_1^{(T)}||). \tag{23}$$

$\square$

(a) We note that $\delta_{(\cdot)}^{\boldsymbol{\mu}} = ||\boldsymbol{\mu}_1^{(\cdot)} - \boldsymbol{\mu}_0^{(\cdot)}||$ and $\Delta_i^{\boldsymbol{\mu}} = ||\boldsymbol{\mu}_i^{(T)} - \boldsymbol{\mu}_i^{(S)}||$.

**Proposition 2** (Bound for $D_{S,T}^\gamma(P;\lambda)$). *For any $\gamma \ge 0$, and under the assumption that the prior distribution $P$ over the classification function family $\mathcal{H}$ is defined, we establish a bound for the domain discrepancy measure $D_{S,T}^{\gamma/2}(P;\lambda)$. Specifically, we have the following inequality:*

$$D_{S,T}^{\gamma/2}(P;\lambda) \le O\left(\sum_{i \in V^S}\sum_{j \in V^T} ||(A^S X^S)_i - (A^T X^T)_j||_2^2 + \sum_{i \in V^S}\sum_{j \in V^T} ||X_i^S - X_j^T||_2^2\right). \tag{24}$$

*Proof.* For notational simplicity, let $h_i \equiv h_i(X,G)$ for any $i \in V_S \cup V_T$. Define $\eta_k(i) = \Pr(y_i = k \mid g_i(X,G))$ for $k \in \{0,1\}$, and let $\mathcal{L}^\gamma(h_i, y_i) = \mathbb{1}\left[h_i[y_i] \le \gamma + \max_{k \ne y_i} h_i[k]\right]$.

We can express the difference in the loss functions as follows:

$$\mathcal{L}_T^{\gamma/2}(h) - \mathcal{L}_S^\gamma(h) = \mathbb{E}_{y^T}\left[\frac{1}{N_T}\sum_{j \in V_T}\mathcal{L}^{\gamma/2}(h_j, y_j)\right] - \mathbb{E}_{y^S}\left[\frac{1}{N_S}\sum_{i \in V_S}\mathcal{L}^\gamma(h_i, y_i)\right] \tag{25}$$

$$\le \frac{1}{\max(N_S, N_T)}\mathbb{E}_{y^S, y^T}\sum_{i \in V_S}\left(\frac{1}{N_T}\sum_{j \in V_T}\mathcal{L}^{\gamma/2}(h_j, y_j) - \mathcal{L}^\gamma(h_i, y_i)\right). \tag{26}$$

Using Definition 1, we derive:

$$\mathcal{L}_T^{\gamma/2}(h) - \mathcal{L}_S^\gamma(h) = \frac{1}{\max(N_S, N_T)}\sum_{i \in V_S}\frac{1}{N_T}\left(\sum_{j \in V_T}\mathbb{E}_{y_j}\mathcal{L}^{\gamma/2}(h_j, y_j) - \mathbb{E}_{y_i}\mathcal{L}^\gamma(h_i, y_i)\right)$$

$$= \frac{1}{\max(N_S, N_T)}\sum_{i \in V_S}\frac{1}{N_T}\sum_{j \in V_T}\sum_k\left(\eta_k(j)\mathcal{L}^{\gamma/2}(h_j, k) - \Pr(y_i = k)\mathcal{L}^\gamma(h_i, k)\right)$$

$$= \frac{1}{\max(N_S, N_T)}\sum_{i \in V_S}\frac{1}{N_T}\sum_{j \in V_T}\sum_k\left(\eta_k(j)\mathcal{L}^{\gamma/2}(h_j, k) - \eta_k(i)\mathcal{L}^\gamma(h_i, k)\right)$$

$$= \frac{1}{\max(N_S, N_T)}\sum_{i \in V_S}\frac{1}{N_T}\sum_{j \in V_T}\sum_k\left(\eta_k(j)\left(\mathcal{L}^{\gamma/2}(h_j, k) - \mathcal{L}^\gamma(h_i, k)\right) + (\eta_k(j) - \eta_k(i))\mathcal{L}^\gamma(h_i, k)\right) \tag{27}$$

$$\le \frac{1}{\max(N_S, N_T)}\sum_{i \in V_S}\frac{1}{N_T}\sum_{j \in V_T}\sum_k\left(\mathcal{L}^{\gamma/2}(h_j, k) - \mathcal{L}^\gamma(h_i, k) + ||\eta_k(j) - \eta_k(i)||_2^2\right). \tag{28}$$

The last inequality holds since both $\eta_k(j)$ and $\mathcal{L}^\gamma(h_i, k)$ are upper-bounded by 1, and we assume $\mathcal{L}^{\gamma/2}(h_j, k) \le \mathcal{L}^\gamma(h_i, k)$.

By applying Theorem 2, we obtain:

$$\sum_k ||\eta_k(j) - \eta_k(i)||_2^2 \le O\left(||\mathbf{f}_u - \mathbf{f}_v||_2^2 + ||\mathbf{f}_v - \boldsymbol{\mu}_1^{(S)}||_2^2 + ||\mathbf{f}_v - \boldsymbol{\mu}_1^{(T)}||_2^2 + ||\mathbf{f}_v - \boldsymbol{\mu}_0^{(S)}||_2^2 + ||\mathbf{f}_v - \boldsymbol{\mu}_0^{(T)}||_2^2\right).$$

Thus, we have:

$$\mathcal{L}_T^{\gamma/2}(h) - \mathcal{L}_S^{\gamma}(h) \leq \frac{1}{\max(N_S, N_T)} \sum_{i \in V_S} \frac{1}{N_T} \sum_{j \in V_T} \sum_k \|\eta_k(j) - \eta_k(i)\|_2^2$$

$$\leq O\left( \sum_{i \in V_S} \sum_{j \in V_T} \|\mathbf{f}_u - \mathbf{f}_v\|_2^2 + \|\mathbf{f}_v - \boldsymbol{\mu}_1^{(S)}\|_2^2 + \|\mathbf{f}_v - \boldsymbol{\mu}_1^{(T)}\|_2^2 + \|\mathbf{f}_v - \boldsymbol{\mu}_0^{(S)}\|_2^2 + \|\mathbf{f}_v - \boldsymbol{\mu}_0^{(T)}\|_2^2 \right)$$

$$\leq O\left( \sum_{i \in V_S} \sum_{j \in V_T} \|(A^S X^S)_i - (A^T X^T)_j\|_2^2 + \sum_{i \in V_S} \sum_{j \in V_T} \|X^S i - X_j^T\|_2^2 \right).$$

$\square$

## B    DEFINITION OF AVERAGE FEATURE VALUE

We hope to quantitatively compare the differences in feature values between topology view and attribute view. Similarly, for the purpose of convenient comparison, we decided to calculate the average of their feature values. Specifically, we first obtain a topology view matrix through topology filtering, multiplying $A$ and $X$ to $\mathcal{F}$. Similarly, we perform attribute filtering by multiplying $\hat{A}$ and $X$ to $\mathcal{F}_f$ to obtain a matrix of attribute view. So our topology average value is $Fetuare_t = \sum_{j=1}^d \sum_{i=1}^N |\mathcal{F}| / (d * N)$ and attribute feature value is $Feature_f = \sum_{j=1}^d \sum_{i=1}^N |\mathcal{F}_f| / (d * N)$.

## C    DESCRIPTION OF ALGORITHM GAA

---

**Algorithm 1:** The proposed algorithm GAA

---

**Input:** Source node feature matrix $X^S$; source original graph adjacency matrix $A^S$; Target node feature matrix $X^T$; Target original graph adjacency matrix $A^T$ source node label matrix $Y^S$; maximum number of iterations $\eta$

Compute the feature graph topological structure $\hat{A}^S$ and $\hat{A}^T$ according to $X^S$ and $X^T$ by running $k$NN algorithm.

**for** $it = 1$ **to** $\eta$ **do**

    $Z^S = GCN(A^S, X^S)$

    $Z_f^S = GCN(\hat{A}^S, X^S)$// embedding of source graph

    $Z^T = GCN(A^T, X^T)$

    $Z_f^T = GCN(\hat{A}^T, X^T)$// embedding of target graph

    $Z^S$ and $Z_f^S$ through cross-view similarity matrix refinement to get $S^S$.

    $Z^T$ and $Z_f^T$ through cross-view similarity matrix refinementto get $S^T$.

    Attribute-Driven domain adaptive between $S^S$ and $S^T$// adaptive in two views

    Domain Adaptive Learning between $Z^S$ and $Z^T$

    $\hat{y}_i^S$ constrained by $y_i^S$ and $\hat{y}_i^T$ constrained by $\hat{y}_i^T$

    Calculate the overall loss with Eq.(15)

    Update all parameters of the framework according to the overall loss

**end**

Predict the labels of target graph nodes based on the trained framework.

**Output:** Classification result $\hat{Y}^T$

---

## D    PARAMETER ANALYSIS

$\alpha$, $\beta$, and $\tau$ are chosen from the set $\{0.005, 0.01, 0.1, 0.5, 1, 5\}$. These values provide flexibility for adjusting the relative importance of different loss terms. $k$ (the number of neighbors for $k$-NN graph construction) is typically $k \in \{1, \cdots, 10\}$ . The optimal value for $k$ depends on the density and

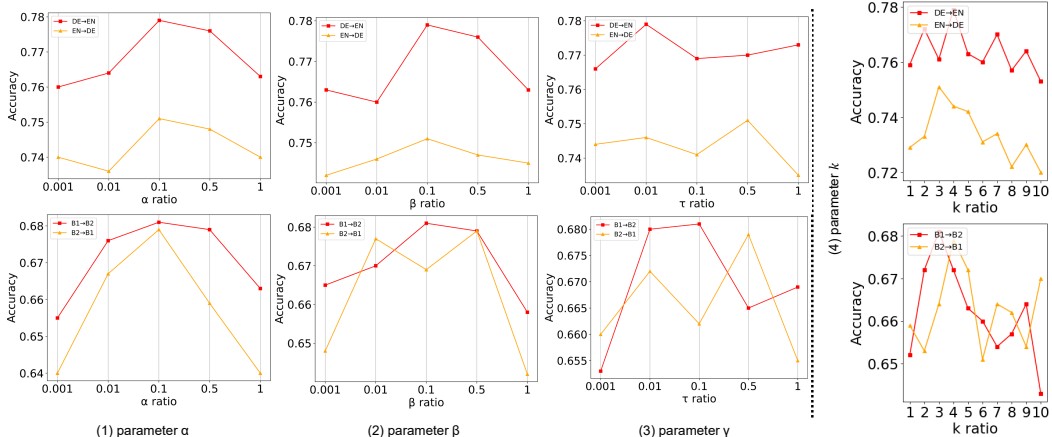

Figure 5: The influence of parameters $\alpha$, $\beta$, $\tau$ and $k$ on two social datasets.

connectivity of the graph. Due to extremely large $k$ could also introduce noisy that will deteriorate the performance. Usually our largest $k$ will be $5$.

Airport Dataset: Often contains transportation networks with fewer nodes but complex edge relationships. Given the sparsity of this dataset, $\alpha$, $\beta$ and $\tau$ should be set relatively higher to emphasize topology alignment and capture key structural relationships. $\alpha$, $\beta$ and $\tau$ is selected from $\{0.1, 0.5\}$. A smaller $k$ could be more effective due to the sparser nature of these networks. We select $k$ from $\{3, 4\}$.

Citation Dataset: This dataset often has a higher node count and diverse structural characteristics. In such datasets, balance the impact of node attributes and topology. $\alpha$, $\beta$ and $\tau$ is selected from $\{0.1, 0.5\}$. A moderate value of $k$ to capture relevant local structures could work well for this dataset. We select $k$ from $\{4, 5\}$.

Social Network Dataset (Blog and Twitch): Social networks often contain a large number of nodes with rich attribute information but high variance in structural patterns. Emphasize attribute alignment since social networks tend to have highly distinctive attributes. Thus, attribute shifts are more sensitive to the values of $\alpha$, $\beta$ and $\tau$, which are selected from the set $\{0.01, 0.1, 0.5\}$. Small $k$ is recommended due to the dense connections in social networks. We select $k$ from $\{3, 4\}$.

MAG Dataset: The MAG dataset is large and diverse, containing losts of classes with various relationships and rich metadata. Structural and attribute alignment are key factors. In this context, attribute shifts are both important to the values of $\alpha$,$\beta$ and $\tau$ which are selected from the set $\{0.1, 0.5\}$. The parameter $k$ works well in this context, enabling the model to capture high-level local and global structural information within the graph. We select $k$ from $\{4, 5\}$.

## E  T-SNE SAMPLE

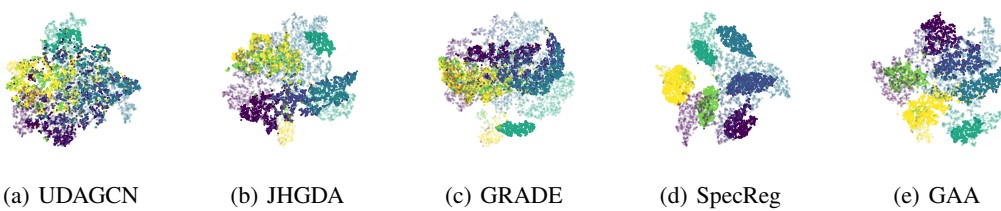

(a) UDAGCN     (b) JHGDA     (c) GRADE     (d) SpecReg     (e) GAA

Figure 6: Visualization of learnt representations of different methods on D-A task of dataset.

| Types | Datasets | $\alpha$ | $\beta$ | $\tau$ | $k$ |
|---|---|---|---|---|---|
| Airport | U→B | 0.5 | 0.5 | 0.01 | 4 |
|  | U→E | 0.1 | 0.1 | 0.01 | 2 |
|  | B→U | 0.1 | 0.1 | 0.01 | 4 |
|  | B→E | 0.5 | 0.1 | 0.1 | 3 |
|  | E→U | 0.5 | 0.5 | 0.1 | 4 |
|  | E→B | 0.5 | 0.5 | 0.1 | 4 |
| Citation | A→D | 0.1 | 0.1 | 0.1 | 3 |
|  | D→A | 0.1 | 0.1 | 0.01 | 4 |
|  | A→C | 0.5 | 0.5 | 0.01 | 4 |
|  | C→A | 0.1 | 0.1 | 0.1 | 3 |
|  | C→D | 0.1 | 0.1 | 0.1 | 4 |
|  | D→C | 0.1 | 0.1 | 0.1 | 4 |
| Blog | B1→B2 | 0.1 | 0.1 | 0.1 | 2 |
|  | B2→B1 | 0.1 | 0.1 | 0.1 | 3 |
| Twitch | DE→EN | 0.1 | 0.1 | 0.01 | 2 |
|  | EN→DE | 0.1 | 0.5 | 0.5 | 2 |
| MAG | US→CN | 0.5 | 0.1 | 0.1 | 5 |
|  | US→DE | 0.1 | 0.1 | 0.1 | 5 |
|  | US→JP | 0.1 | 0.5 | 0.01 | 6 |
|  | US→RU | 0.1 | 0.1 | 0.5 | 5 |
|  | US→FR | 0.1 | 0.1 | 0.1 | 6 |
|  | CN→US | 0.1 | 0.1 | 0.01 | 6 |
|  | CN→DE | 0.1 | 0.1 | 0.5 | 6 |
|  | CN→JP | 0.1 | 0.1 | 0.01 | 5 |
|  | CN→RU | 0.5 | 0.1 | 0.1 | 5 |
|  | CN→FR | 0.1 | 0.01 | 0.1 | 6 |

Table 5: Experiment hyperparameter setting Value.

## F  TIGHTNESS OF BOUNDS

To evaluate the tightness of our bounds, we conduct additional experiments to verify the effects of node attribute divergence and topology divergence independently. The following experimental detail settings are designed to verify these divergences.

### F.1  ATTRIBUTE DIVERGENCE

To evaluate the impact of graph attribute discrepancy on GDA, we designed an experiment for this purpose. In this experiment, we provide node classification tasks across different graphs under different attribute discrepancy with same topology structure. In this procedure, we aim to generate a collection of graph datasets, where each graph is characterized by a fixed adjacency matrix $A$, consisting of 100 nodes with an average degree of 0.3, and node attribute matrices $X$ randomly simulated from Gaussian-distributed samples. The specific steps are as follows: Each graph $G_i = (A, X_i)$ shares the same fixed adjacency matrix $A$, representing the same graph topology. $A$ is predetermined and defines the connectivity between nodes, remaining consistent across all generated graphs. Node attributes $X_i$ for each graph $G_i$ are generated using 'make-blobs' function from scikit-learn. Parameters for 'make-blobs': Number of nodes: $n_{\text{samples}} = 100$, representing the total number of nodes in the graph. Number of clusters: centers $= 2$, corresponding to two distinct classes. $n_{\text{features}} = 10$, meaning each node is described by a 10-dimensional feature vector. cluster_std is a variable parameter uniformly sampled from the range $[0, 2]$, determining the dispersion of node features within each cluster. We construct a dataset of 1000 graphs, $\{G_i = (A, X_i)\}_{i=1}^{1000}$, where: $A$: the adjacency matrix, remains fixed across all graphs, representing the structural relationships between nodes. $X_i$: the feature matrix, varies between graphs. The variance of the node features is determined by cluster_std, which is uniformly sampled for each graph to introduce diversity in the node attributes. GAA is trained for 100 epochs on a fixed source graph and target graphs with different attribute variances. After training, we reports three key metrics for each dataset: the bound value, $\mathcal{L}_A$ (loss value), and the target graph accuracy. To ensure that the bound value and loss value are on the same scale, we normalize the bound value by dividing it by the number of nodes, i.e., 100. As illustrated in Figure 6(a), both the bound value and the loss value of the model increase as

the attribute discrepancy grows. Conversely, the classification performance declines with increasing attribute discrepancy, highlighting that the bound attribute component is closely related to the GDA performance.

### F.2 TOPOLOGY DIVERGENCE

This procedure involves generating 1000 graphs by Stochastic Block Model (SBM), a probabilistic model for community-structured graphs. Each graph consists of different adjacency matrix $A$ with uniformly distributed edge weights and a node attribute matrix $X$ with fixed-dimensional feature vectors. The generation process is detailed below: The graph $G_i = (A_i, X_i)$ for each instance is generated using the SBM. SBM parameters are as follows: community contain 100 nodes( num_nodes = 100): sizes $= \left[ \frac{\text{num\_nodes}}{2}, \text{num\_nodes} - \frac{\text{num\_nodes}}{2} \right]$, where the graph is divided into two communities of approximately equal size, which can be seen as 2 classes. Inter- and Intra-community connection probabilities: probs $= \begin{bmatrix} p & \frac{p}{10} \\ \frac{p}{10} & p \end{bmatrix}$, where $p = 0.8$ denotes the probability of edges forming within a community and $\frac{p}{10} = 0.08$ denotes the probability of edges forming different communities. To incorporate variability in edge strengths, the weights of edges in the adjacency matrix $A_i$ are drawn from a uniform distribution. Nonexistent edges are assigned a weight of $0$, thereby preserving the sparsity structure dictated by the Stochastic Block Model (SBM). Each adjacency matrix $A_i$, representing graph $G_i$, is a symmetric $n \times n$ matrix. Additionally, each graph $G_i$ is associated with a node attribute matrix $X_i$, where $X_i$ contains $n$ rows, corresponding to the attributes of the $n$ nodes. Each nodes is a fixed 10-dimensional vector, ensuring consistent node attribute dimensionality across all graphs. All elements of $X_i$ are set to a constant value of $1$, ensuring uniform node attributes across the dataset. The dataset comprises 1000 graphs, each containing 100 nodes, represented as $\{G_i = (A_i, X_i)\}_{i=1}^{1000}$. In this representation: $A_i$ varies between graphs, following the Stochastic Block Model (SBM) with uniform edge weights, while $X_i$ is a fixed matrix where each of its 10-dimensional elements is set to $1$. GAA is trained for 100 epochs on a fixed source graph and target graphs with different topology variances. After training, we reports three key metrics for each dataset: the bound value, $\mathcal{L}_A$ (loss value), and the target graph accuracy. Similarly, we normalize the bound value by dividing it by the number of nodes, i.e., $100$. As illustrated in Figure 6(b), both the bound value and the loss value of the model increase as the attribute discrepancy grows. Conversely, the classification performance declines with increasing topology discrepancy, emphasizing that the bound's topology component is also closely linked to the GDA performance.

## G MODEL EFFICIENT EXPERIMENT

To further investigate the efficiency of GAAo, Table**??** reports the running time comparison across various algorithms. We also compared the training time and GPU memory usage of common baselines UDAGCN and a recent SOTA method, JHGDA, which aligns graph domain discrepancy hierarchical levels. As shown in Table, the evaluation results on airport dataset further demonstrate that our method achieves superior performance with tolerable computational and storage overhead.

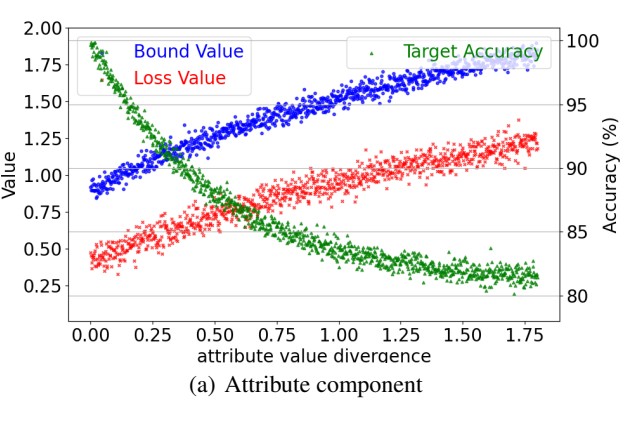

(a) Attribute component

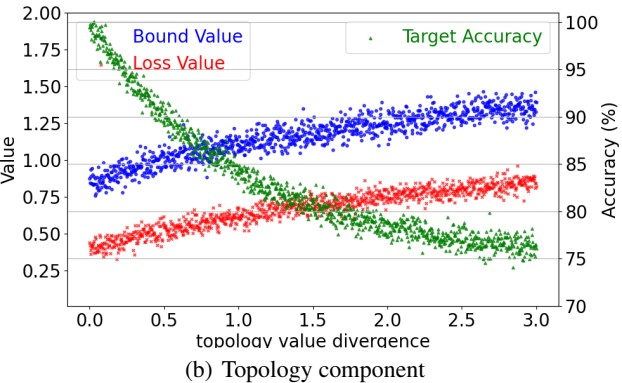

(b) Topology component

Figure 7: Visualization of bound value and $\mathcal{L}_A$ value.

| Dataset | Method | Training Time (Normalized w.r.t. UDAGCN) | Memory Usage (Normalized w.r.t. UDAGCN) | Accuracy(%) |
|---|---|---|---|---|
| U→B | UDAGCNB | 1 | 1 | 0.607 |
| | JHGDA | 1.314 | 1.414 | 0.695 |
| | PA | **0.498** | 0.517 | 0.679 |
| | $GAA_o$ | 0.504 | **0.514** | 0.697 |
| | $GAA$ | 1.063 | 1.113 | **0.704** |
| U→E | UDAGCNB | 1 | 1 | 0.488 |
| | JHGDA | 1.423 | 1.513 | 0.519 |
| | PA | 0.511 | **0.509** | 0.557 |
| | $GAA_o$ | **0.507** | 0.513 | 0.556 |
| | $GAA$ | 1.109 | 1.098 | **0.563** |
| B→E | UDAGCNB | 1 | 1 | 0.510 |
| | JHGDA | 1.311 | 1.501 | 0.569 |
| | PA | **0.502** | **0.497** | 0.562 |
| | $GAA_o$ | 0.507 | 0.503 | 0.566 |
| | $GAA$ | 1.048 | 1.107 | **0.573** |

Table 6: Comparison of Training Time, Memory Usage, and Accuracy on Airport datset.

