# OpenReview forum: "On the Benefits of Attribute-Driven Graph Domain Adaptation"
_ICLR.cc/2025/Conference — ICLR 2025 Poster_

### Official Review · Reviewer_KMc6 · 2024-11-01

**Soundness:** 3
**Presentation:** 3
**Contribution:** 3
**Rating:** 6
**Confidence:** 4

**Summary:**

In this paper, we focus on the graph domain adaptation (GDA) problem and propose a novel cross-channel graph attribute-driven Alignment (GAA) algorithm, which aims to address the distribution differences between the source and target graphs, especially the differences in node attributes and topology.

**Strengths:**

1. Through theoretical and experimental analysis, the difference of eigenvalues distribution between graph topology and node attributes is analyzed, which reveals that the difference of node attributes is more significant than the difference of topology structure, and further emphasizes the importance of node attribute alignment.
2. The Cross-Channel Module is proposed to align the graph topology and node attributes simultaneously. This module effectively reduces the domain differences by fusing and aligning the two views of the source and target images.

**Weaknesses:**

1. The whole setting of this paper is to perform domain adaptation (GDA) under supervised conditions, that is, the annotation of the target domain is involved in the training process. However, for most practical problems, the target domain is extremely lack of data annotation or even no annotation, which lacks certain considerations for practical applications. If the target domain is annotated why not design the model for the target domain instead of introducing the source domain.
2. In the experiments section, we see that the latest method you currently cite is the result of 2023, and the comparison test of the relevant latest method in 2024 is missing. And on the types of downstream tasks, there is a lack of evaluation of other tasks, such as graph classification tasks.
3. This paper lacks the time complexity analysis of the model, and does not analyze the overall operating efficiency of the model. In particular, GAA algorithm introduces additional calculation steps (such as cross-view similarity matrix), which may lead to increased computational complexity. On large-scale graph data, the efficiency and scalability of the algorithm may become an issue.
4. In the model graph part of this paper, in the target domain part, the legend labels of the attribute graph and the topology graph are wrong. The text annotation at the bottom of Figure 1 is ambiguous from the data in the bar graph (the difference in the distribution of eigenvalues is greater for topological graphs than for attributed graphs?).

**Questions:**

1. This paper does not analyze the time complexity of the model, whether it is universal for larger scale node classification problems, and whether it can ensure the operation efficiency of the model on large-scale problems. For example, in the construction of attributed graph using KNN, is it too time-consuming to successively calculate the similarity score between nodes to connect edges?
2. The article is generally on the supervised domain adaptation problem. Personally, I think this application scenario is relatively limited in the general scenario.
3. The article does not give detailed hyperparameter Settings, for example, in the weight design of the loss function, although the experiment carries out the analysis of different weights, the final setting value is not given. In addition, for the weight setting of the loss function, will the weight of a loss function be too low to cause it to not work in the model training process?
4. Can it be compared with the latest GDA method?

---

> ### Author Response · Authors · 2024-11-23
> **Reply from the Authors**
>
> **Part 1/3**
>
> Thank you for your constructive feedback! Below, we address the concerns and questions raised in the weaknesses section. Please feel free to reach out if further clarification is required.
>
> ## Q1
>
> We introduce Cross-view Similarity Matrix and Attention-based Attribute components to further boost the performance of GAA. This is essentially a **tradeoff between algorithm performance and computational complexity**. In light of your comments, we conducted additional experiments by removing the Cross-view Similarity Matrix and Attention-based Attribute, and the results are shown in **Table 1-1** and **Table 1-2**. It can be observed that there is a slight performance drop after removing the Cross-view Similarity Matrix and Attention-based Attribute component. Specifically, we replace $att^S$ with $Z^S$ , $att^T$ with $Z^T$ , $att_f^S$ with $Z_f^S$ , and $att_f^T$ with $Z_f^T$. However, we note that it still outperforms other SOTA methods, such as recent 2024 work PA[1], which highlights the importance of attribute alignment in GAA.
>
> | Methods | U $\rightarrow$ B | U $\rightarrow$ E | B $\rightarrow$ U | B $\rightarrow$ E | E $\rightarrow$ U | E $\rightarrow$ B | DE $\rightarrow$ EN | EN $\rightarrow$ DE |
> | --- | --- | --- | --- | --- | --- | --- | --- | --- |
> | **PA** | 0.679 | $\underline{\text{0.557}}$ | 0.528 | 0.562 | 0.532 | 0.529 | 0.677 | 0.739 |
> | **GAA_o** | $\underline{\text{0.697}}$ | 0.552 | $\underline{\text{0.537}}$ | $\underline{\text{0.569}}$ | $\underline{\text{0.538}}$ | $\underline{\text{0.682}}$ | $\underline{\text{0.773}}$ | $\underline{\text{0.746}}$ |
> | GAA | **0.704** | **0.563** | **0.542** | **0.573** | **0.546** | **0.691** | **0.779** | **0.751** |
>
> **Table 1-1.** Cross-network node classification on the airport network and social network.
>
> | Methods | A $\rightarrow$ D | D $\rightarrow$ A | A$ \rightarrow$ C | C $\rightarrow$ A | C $\rightarrow$ D | D $ \rightarrow$ C | B1 $\rightarrow$ B2 | B2 $\rightarrow$ B1 |
> | --- | --- | --- | --- | --- | --- | --- | --- | --- |
> | **PA** | 0.752 | 0.740 | $\underline{\text{0.817}}$ | 0.768 | 0.755 | 0.780 | 0.662 | 0.654 |
> | **GAA_o** | $\underline{\text{0.773}}$ | $\underline{\text{0.749}}$ | 0.815 | $\underline{\text{0.775}}$ | $\underline{\text{0.764}}$ | $\underline{\text{0.792}}$ | $\underline{\text{0.675}}$ | $\underline{\text{0.662}}$ |
> | GAA | **0.789** | **0.754** | **0.824** | **0.782** | **0.771** | **0.798** | **0.681** | **0.679** |
>
> **Table1-2.** Cross-network node classification on the citation, blog network.
>
> To further investigate the efficiency of **GAAo**, **Table 2** reports the running time comparison across various algorithms. We also compared the training time and GPU memory usage of common baselines UDAGCN [2] and a recent SOTA method, JHGDA, which aligns graph domain discrepancy hierarchical levels [3]. As shown in **Table 2**, the evaluation results on **U $\rightarrow$ B** further demonstrate that, with tolerable computational and storage overhead, our method achieves superior performance.
>
> | Methods | Training Time (Normalized w.r.t. UDAGCN) | Memory Usage (Normalized w.r.t. UDAGCN) | Accuracy (%) |
> | --- | --- | --- | --- |
> | **UDAGCN** | 1   | 1   | 0.607 |
> | **JHGDA** | 1.314 | 1.414 | 0.695 |
> | **PA** | **0.498** | $\underline{\text{0.517}}$ | 0.679 |
> | **GAA_o** | $\underline{\text{0.504}}$ | **0.514** | $\underline{\text{0.697}}$ |
> | **GAA** | 1.063 | 1.113 | **0.704** |
>
> **Table2.** Comparison of Training Time, Memory Usage, and Accuracy on **U $\rightarrow$ B**.
>
> We also conduct one additional experiment on large-scale data MAG, and the results are shown in **Table 3**. It can be observed that GAA achieves state-of-the-art (SOTA) performance on large-scale datasets. Details of other baselines and dataset information are provided in the revised version (**Table 4** and **Table 1**).
>
> | Methods | US $\rightarrow$ CN | US $\rightarrow$ DE | US $\rightarrow$ JP | US $\rightarrow$ RU | US $\rightarrow$ FR | CN $\rightarrow$ US | CN $\rightarrow$ DE | CN $\rightarrow$ JP | CN$ \rightarrow$ RU | CN$ \rightarrow $FR |
> | --- | --- | --- | --- | --- | --- | --- | --- | --- | --- | --- |
> | **PA** | 0.400 | 0.389 | 0.474 | $\underline{\text{0.371}}$ | 0.252 | $\underline{\text{0.452}}$ | 0.262 | 0.383 | 0.333 | 0.242 |
> | **GAA_o** | $\underline{\text{0.402}}$ | $\underline{\text{0.394}}$ | $\underline{\text{0.481}}$ | 0.363 | $\underline{\text{0.279}}$ | 0.445 | $\underline{\text{0.293}}$ | $\underline{\text{0.395}}$ | $\underline{\text{0.348}}$ | $\underline{\text{0.272}}$ |
> | GAA | **0.410** | **0.401** | **0.492** | **0.372** | **0.288** | **0.453** | **0.302** | **0.400** | **0.351** | **0.293** |
>
> **Table 3.** Cross-network node classification on MAG datasets.

---

> ### Author Response · Authors · 2024-11-23
> **Reply from the Authors**
>
> **Part 2/3**
>
> Regarding $k$NN, we follow existing works in the literature e.g., [4, 5, 6], which adopt it as a pre-processing step to construct the attribute view. Specifically, we note that the construction of attribute view using $k$NN is **tied to the graph dataset itself**, but does not rely on any specific algorithms or downstream tasks (i.e., we can obtain the attribute view prior to implementing our algorithm). In other words, it does not affect the computational complexity of the algorithm.
>
>
> ## Q2
>
> We would like to clarify that our paper specifically addresses the **unsupervised domain adaptation (UDA)** problem, where the model adapts knowledge from a source domain (with labeled data) to a target domain (without labeled data in **line 161** target graph $G^T$) . Additionally, our target graph classification loss $\mathcal{L_T}$​ (**Section 4.4, Eq. (4), line 339**) does not rely on labels. $\hat{y}_i^T$​ is a pseudo-label that is obtained through dimensionality reduction. GAA focuses on aligning the **node attributes** and **topological structures** between the source and target graphs in the absence of labeled target data, which is a critical challenge in many real-world scenarios.
>
> We believe that this unsupervised setting is highly important, as labeled data in the target domain is often unavailable or expensive to obtain. While our method does not rely on target labels, it employs novel techniques such as attribute alignment to achieve effective adaptation.
>
> ## Q3
>
> To address your concern, we have provided a detailed explanation of our hyperparameter selection strategy. Additionally, we discuss the optimal parameter settings in the context of our study. Furthermore, a comprehensive overview of the **hyperparameter setting value** for different datasets is provided in the revised manuscript **Appendix D, Table 5**.
>
> $\alpha$, $\beta$, and $\tau$ are chosen from the set $\{0.005, 0.01, 0.1, 0.5, 1, 5\}$. These values provide flexibility for adjusting the relative importance of different loss terms. $k $(the number of neighbors for $k$-NN graph construction) is typically $k \in \[1, \cdots, 10 \]$ . The optimal value for k depends on the density and connectivity of the graph. The extremely large $k$ could also introduce noise that will deteriorate the performance [5]. Usually, our largest $k$ will be $6$ [4].
>
> **Airport Dataset**: Often contains transportation networks with fewer nodes but complex edge relationships. Given the complexity of this dataset, $\alpha$, $\beta$, and $\tau$ should be set relatively higher to emphasize topology alignment and capture key structural relationships. $\alpha$, $\beta$ and $\tau$ is selected from $\{ 0.1, 0.5 \}$. A wide $k$ could be more effective due to the nature of these networks (containing both density and sparse). We select $k$ from $\{ 2, 3, 4\}$.
>
> **Citation Dataset**: This dataset often has a higher node count and diverse structural characteristics. In such datasets, balancing the impact of node attributes and topology is important. $\alpha$, $\beta$ and $\tau$ is selected from $\{ 0.1, 0.5 \}$. A middle value of $k$ to capture relevant local structures could work well for this dataset. We select $k$ from $\{ 3, 4 \}$.
>
> **Social Network Dataset (Blog and Twitch)**: Social networks often contain a large number of nodes with rich attribute information but high variance in topology structure. Emphasize attribute consistency since social networks have distinctive attributes. Thus, attribute discrepancy is more sensitive to the values of $\alpha$, $\beta$, and $\tau$, which are selected from the set $\{0.01, 0.1, 0.5\}$. Small $k$ is recommended due to the dense connections in social networks. We select $k$ from $\{ 2, 3 \}$.
>
> **MAG Dataset**: The MAG dataset is large and diverse, containing lots of classes with various relationships and rich metadata. Structural and attribute alignment are key factors. In this context, attribute shifts are both important to the values of $\alpha$, $\beta$, and $\tau$, which are selected from the set $\{0.1, 0.5\}$. The parameter $k$ works well when enabling the model to capture high-level local and global structural information within the graph. We select $k$ from $\{ 5, 6 \}$.
>
> In summary, the weight of the loss function remains stable between 0.1 and 0.5, indicating that each loss function works well.

---

> ### Author Response · Authors · 2024-11-23
> **Reply from the Authors**
>
> **Part 3/3**
>
> ## Q4
>
> To address your concern, we have included a comparison with the two latest state-of-the-art (SOTA) methods from 2024( **Table 1-1**, **Table 1-2** and **Table 3**). Additionally, detailed descriptions of the experimental settings and an expanded comparison with other SOTA methods can be found in the revised manuscript ( **Section 5.4, Table 2, Table 3, Table 4**）.
>
> [1] Liu, Shikun, et al. "Pairwise Alignment Improves Graph Domain Adaptation." *International Conference on Machine Learning*, 2024.
>
> [2] Man, Wu, et al. "Unsupervised Domain Adaptive Graph Convolutional Networks." *The Web Conference*, 2020.
>
> [3] Shi, Boshen, et al. "Improving Graph Domain Adaptation with Network Hierarchy." *Conference on Information and Knowledge Management*, 2023.
>
> [4] Chen, Jie, et al. "Fast Approximate kNN Graph Construction for High Dimensional Data via Recursive Lanczos Bisection." *Journal of Machine Learning Research*, 2009.
>
> [5] Wang, Xiao, et al. "AM-GCN: Adaptive Multi-channel Graph Convolutional Networks." *SIGKDD International conference on knowledge discovery & data mining*, 2020.
>
> [6] Fang, Ruiyi, et al. "Structure-Preserving Graph Representation Learning." *International Conference on Data Mining*, 2022.

---

> > ### Comment · Reviewer_KMc6 · 2024-11-25
> >
> > Thank you for your feedback, which effectively addressed my concern and I am willing to improve my score.

---

> ### Author Response · Authors · 2024-11-25
> **Thank you for your feedback!**
>
> Dear Reviewer,
>
> Thank you for your thoughtful feedback and for increasing the score of our manuscript. We sincerely appreciate your insightful questions, as addressing them has significantly strengthened our paper. We welcome any additional questions or suggestions you may have.
> Sincerely,
>
> The Authors

---

### Official Review · Reviewer_JjmB · 2024-11-01

**Soundness:** 3
**Presentation:** 3
**Contribution:** 2
**Rating:** 6
**Confidence:** 3

**Summary:**

This paper studies the problem of Graph Domain Adaptation and shows the significance of the graph node attribute for graph domain alignment. Then, they propose a novel cross-channel module to fuse and align both views between the source and target graphs for GDA. Experimental results on a variety of benchmarks verify the effectiveness of the method.

**Strengths:**

- The studied problem is interesting and important.
- The paper is well-organized and clearly written.
- The experiments are extensive and helpful to validate the effectiveness of the model.

**Weaknesses:**

- Some citation styles are not correct, e.g., Line 37.
- The paper lacks some of the latest baselines such as "Information filtering and interpolating for semi-supervised graph domain adaptation (2024)"
- Masking node attributes could have the risk of losing crucial information. How about alignment in the embedding space?
- I suggest that the authors should use some tSNE to show the effectiveness of the method.
- The parameter sensitivity needs more analysis. How to apply your method to a new dataset.

**Questions:**

See above.

---

> ### Author Response · Authors · 2024-11-23
> **Reply from the Authors**
>
> **Part 1/2**
>
> Thank you for your appreciated feedback. Below, we address the concerns and questions raised in the weaknesses section. Please feel free to reach out if further clarification is required.
>
> ## Q1
>
> Thank you for your comment regarding the citation style. We apologize for the inconsistencies in our citations and have carefully reviewed and corrected them throughout the paper.
>
> ## Q2
>
> We acknowledge the importance of comparing our approach with recent advancements, including GIFI [1]. Although our method focuses on unsupervised graph domain adaptation, we recognize the value of benchmarking against semi-supervised methods for broader context. In our revised submission, we have incorporated this baseline into our experiments (see **Section 5.4**, **Table 3** and **Table 4**). Our results demonstrate that while their method performs well in semi-supervised settings, our approach exhibits outperformance in unsupervised scenarios, highlighting its broader applicability. Additionally, we have clarified the distinctions between the two settings in **Section 5.2**.
>
> | Methods | U $\rightarrow$ B | U $\rightarrow$ E | B $\rightarrow$ U | B $\rightarrow$ E | E $\rightarrow$ U | E $\rightarrow$ B | DE $\rightarrow$ EN | EN $\rightarrow$ DE |
> | --- | --- | --- | --- | --- | --- | --- | --- | --- |
> | **GIFI** | 0.636 | 0.521 | 0.493 | 0.535 | 0.501 | 0.623 | 0.719 | 0.705 |
> | GAA | **0.704** | **0.563** | **0.542** | **0.573** | **0.546** | **0.691** | **0.779** | **0.751** |
>
> **Table1-1.** Cross-network node classification on the airport network and social network.
>
> | Methods | A $\rightarrow$ D | D $\rightarrow$ A | A $\rightarrow$ C | C $\rightarrow$ A | C $\rightarrow$ D | D $\rightarrow$ C | B1 $\rightarrow$ B2 | B2 $\rightarrow$ B1 |
> | --- | --- | --- | --- | --- | --- | --- | --- | --- |
> | **GIFI** | 0.751 | 0.737 | 0.793 | 0.755 | 0.739 | 0.751 | 0.653 | 0.642 |
> | GAA | **0.789** | **0.754** | **0.824** | **0.782** | **0.771** | **0.798** | **0.681** | **0.679** |
>
> **Table1-1.** Cross-network node classification on the Citation, Blog network.
>
> ## Q3
>
> Regarding the concern about masking node attributes and potential information loss, we would like to clarify that our method does **not utilize node attributes masking**. Our approach leverages full node attributes to preserve all available information. Although GAA is not designed to align in the embedding space, our approach achieves alignment by technically aligning graph node attributes and topology discrepancy in the embedding space. This ensures that representations in the embedding space are well-suited for domain adaptation without masking. We acknowledge the importance of exploring alignment in the embedding space as an independent strategy. This aligns with your suggestion and provides a promising avenue for future research. We will consider incorporating dedicated embedding alignment techniques in follow-up studies to further validate and improve our method.
>
> ## Q4
>
> We are pleased to provide additional T-SNE visualizations of citation datasets to demonstrate the effectiveness of our method. Detailed visualization results are in **Appendix F** of our updated manuscript. We observe from **Appendix F** compared with other SOTA methods. Firstly, nodes belonging to different classes are well-separated, demonstrating that GAA effectively distinguishes between classes in the embedding space. Secondly, nodes from the same class across different domains largely overlap, indicating that GAA significantly reduces domain discrepancies. The first observation highlights strong classification performance, while the second suggests effective domain adaptation.

---

> ### Author Response · Authors · 2024-11-23
> **Reply from the Authors**
>
> **Part 2/2**
>
> ## Q5
>
> Following your suggestion, we also conduct one additional experiment on the hyperparameter analysis experiment in our revised manuscript **Appendix D, Figure 5**
>
> To address your concern further, we have provided a detailed explanation of our hyperparameter selection strategy and discussed the optimal parameter settings in the context of our study. Furthermore, the revised manuscript includes a comprehensive overview of the hyperparameter selection process for different datasets, as outlined in **Appendix D, Table 5**.
>
> $\alpha$, $\beta$, and $\tau$ are chosen from the set $\{0.005, 0.01, 0.1, 0.5, 1, 5\}$. These values provide flexibility for adjusting the relative importance of different loss terms. $k$ (the number of neighbors for $k$-NN graph construction) is typically $k \in \[1, \cdots, 10 \]$ . The optimal value for $k$ depends on the density and connectivity of the graph. The extremely large $k$ could also introduce noise that will deteriorate the performance [5]. Usually, our largest $k$ will be $6$ [4].
>
> **Airport Dataset**: Often contains transportation networks with fewer nodes but complex edge relationships. Given the complexity of this dataset, $\alpha$, $\beta$, and $\tau$ should be set relatively higher to emphasize topology alignment and capture key structural relationships. $\alpha$, $\beta$ and $\tau$ is selected from $\{ 0.1, 0.5 \}$. A wide $k$ could be more effective due to the nature of these networks (containing both density and sparse). We select $k$ from $\{ 2, 3, 4\}$.
>
> **Citation Dataset**: This dataset often has a higher node count and diverse structural characteristics. Balancing the impact of node attributes and topology is important in such datasets. $\alpha$, $\beta$ and $\tau$ is selected from $\{ 0.1, 0.5 \}$. A middle value of $k$ to capture relevant local structures could work well for this dataset. We select k from $\{ 3, 4 \}$.
>
> **Social Network Dataset (Blog and Twitch)**: Social networks often contain a large number of nodes with rich attribute information but high variance in topology structure. Emphasize attribute consistency since social networks have distinctive attributes. Thus, attribute discrepancy is more sensitive to the values of $\alpha$, $\beta$, and $\tau$, which are selected from the set $\{0.01, 0.1, 0.5\}$. Small $k$ is recommended due to the dense connections in social networks. We select $k$ from $\{ 2, 3 \}$.
>
> **MAG Dataset**: The MAG dataset is large and diverse, containing lots of classes with various relationships and rich metadata. Structural and attribute alignment are key factors. In this context, attribute shifts are both important to the values of $\alpha$, $\beta$, and $\tau$, which are selected from the set $\{0.1, 0.5\}$. The parameter k works well when enabling the model to capture high-level local and global structural information within the graph. We select $k$ from $\{ 5, 6 \}$.
>
> [1] Qiao, Ziyue, et al. "Information filtering and interpolating for semi-supervised graph domain adaptation." *Pattern Recognition*, 2024

---

> > ### Comment · Reviewer_JjmB · 2024-11-23
> > **Thanks for your response.**
> >
> > Thanks for your response and I support the paper.

---

> > > ### Author Response · Authors · 2024-11-24
> > > **Thank you for your feedback!**
> > >
> > > Dear Reviewer,
> > >
> > > We thank the reviewer for their positive and constructive feedback. We are glad to know that the reviewer liked our paper. We provide answers to the reviewers' comments below and refer to them in red in our revised manuscript. We hope we were able to provide satisfactory answers. Please feel free to ask if the reviewer requires any further clarification.
> > >
> > > Sincerely,
> > >
> > > The Authors

---

### Official Review · Reviewer_v8TX · 2024-11-08

**Soundness:** 3
**Presentation:** 2
**Contribution:** 3
**Rating:** 6
**Confidence:** 3

**Summary:**

This paper aims to address the Graph Domain Adaptation problem by incorporating both structural information and node attribute values, the latter of which is often neglected by previous approaches. The paper first presents the significance of considering node attribute shifts in GDA. It then derives a PAC-Bayesian analysis, which provides an upper bound on the overall discrepancy in expected loss. This bound relies solely on the divergence between the topologies and node attributes of the source and target graphs. The proposed method combines the topological and node feature representation through an attention module and performs domain adaptive learning between the source and target representations. Experiments showed good performance of the proposed algorithm, compared with multiple baselines.

**Strengths:**

1. This paper makes a good observation that node attributes are important for domain adaptation tasks and then follows up with a theoretical analysis.
2. The authors conducted comprehensive experiments to validate the algorithm's performance with a sufficient number of baseline methods compared.
3. The ablation study shows the effectiveness of the proposed method and its key components.

**Weaknesses:**

1. **Writing**: The writing of the paper can be improved. In particular, citation errors appear on the very first page of the paper. The author should proofread the paper more carefully before submission. Some sentences contain grammatical errors.
2. **Theoretical novelty**: The theoretical analysis of the paper is a direct application of the PAC-Bayesian results derived from [1], including the problem formulation and Theorem 1. The discrepancy between subgroups in [1] can be naturally applied to graphs in the source/target domain.
3. **Tightness of bounds**: The upper bound derived in Proposition 1 is novel, yet I am curious if the upper bound is tight enough to be useful. It would be great if the author could provide some numerical evidence showing the tightness of the bound or experiments that show how the improvement of performance relates to the divergence of both topological structure and data features over training.
4. **Scalability**: The scalability of the proposed algorithm is a concern, especially when applied to large graphs. The GAA algorithm requires constructing and processing both topology and attribute views, with a cross-view similarity matrix that grows with the size of the graph. This can lead to high computational costs, particularly when both views are combined through an attention module. Running time analysis and comparison with existing baselines are needed.

**References**
[1] Ma, J et al. Subgroup generalization and fairness of graph neural networks.

**Questions:**

Please refer to the weakness section.

---

> ### Author Response · Authors · 2024-11-23
> **Reply from the Authors**
>
> **Part 1/2**
>
> Thank you for your constructive feedback! Below, we address the concerns and questions raised in the weaknesses section. Please feel free to reach out if further clarification is required.
>
> ## Q1
>
> Thank you for your comment regarding the citation style. We apologize for the inconsistencies in our citations and have carefully reviewed and corrected this throughout the paper.
>
> ## Q2
>
> As mentioned in **Line 187** of our manuscript, Theorem 1 indeed follows (Ma et al.,2021). On the other hand, we would like to note that Theorem 1 serves merely as a stepping stone for our theoretical analysis; our ultimate goal is to introduce Proposition 1, which highlights the role of attribute and topology divergence in GDA. On the algorithmic side, GAA minimizes both attribute and topology distribution shifts based on intrinsic graph property.
>
> ## Q3
>
> We appreciate your question, which gives us the opportunity to further validate the rationale behind our theoretical analysis and algorithm. Specifically, to investigate how the performance relates to the divergence, we have conducted an additional experiment, where we randomly generated 1000 pairs of graphs (source and target) varying attribute and topology values. At each time, the divergence of attribute and topology values between the source and target graphs was adjusted by utilizing make-blobs and stochastic block model (Details in **Appendix F Section F1, Section F2**). **Figure 7 in Appendix F** presents the normalized bound value divergence (calculated using Eq. (4) from **Proposition 1**) and loss value divergence $\mathcal{L}_A$ (defined in Eq. (12) on page 6) for both attribute and topology features as a function value divergence. It is evident that the theoretical values (blue dots) and actual loss values (red dots) exhibit highly consistent trends.
>
> Furthermore, in each simulation, we first train a model using source values (either attribute or topology) and then evaluate its accuracy on the target graph (**target accuracy**). As shown in Figure 6, we observe that regardless of whether attributes or topology values are used, the target accuracy (green dots) consistently decreases as the divergence between the source and target domains increases. These empirical results highlight that both factors significantly influence the model's performance on the target domain, while the importance of attributes has long been overlooked in existing GDA studies. This further validates the rationale behind our work.

---

> ### Author Response · Authors · 2024-11-23
> **Reply from the Authors**
>
> **Part 2/2**
>
> ## Q4
>
> We introduce Cross-view Similarity Matrix and Attention-based Attribute components to further boost the performance of GAA. This is essentially a **tradeoff between algorithm performance and computional complextity**. In light of your comments, we conduct additional experiments by removing Cross-view Similarity Matrix and Attention-based Attribute and the results are shown in **Table 1-1** and **Table 1-2**. It can be observed that there is a slight performacne drop after removing the Cross-view Similarity Matrix and Attention-based Attribute component. Specifically, we replace $ att^S $ with $ Z^S $, $ att^T $ with $ Z^T $, $ att_f^S $ with $ Z_f^S $, and $ att_f^T $ with $ Z_f^T $. However, we note that it still outperforms other SOTA methods, such as recent 2024 work PA[1], which highlights the importance of attribute alignment in GAA.
>
> | Methods | U $\rightarrow$ B | U $\rightarrow$ E | B $\rightarrow$ U | B $\rightarrow$ E | E $\rightarrow$ U | E $\rightarrow$ B | DE $\rightarrow$ EN | EN $\rightarrow$ DE |
> | --- | --- | --- | --- | --- | --- | --- | --- | --- |
> | **PA** | 0.679 | $\underline{\text{0.557}}$ | 0.528 | 0.562 | 0.532 | 0.529 | 0.677 | 0.739 |
> | **GAA$_o$** | $\underline{\text{0.697}}$ | 0.552 | $\underline{\text{0.537}}$ | $\underline{\text{0.569}}$ | $\underline{\text{0.538}}$ | $\underline{\text{0.682}}$ | $\underline{\text{0.773}}$ | $\underline{\text{0.746}}$ |
> | GAA | **0.704** | **0.563** | **0.542** | **0.573** | **0.546** | **0.691** | **0.779** | **0.751** |
>
> **Table1-1.** Cross-network node classification on the airport network and social network.
>
> | Methods | A $\rightarrow$ D | D $\rightarrow$ A | A $\rightarrow$ C | C $\rightarrow$ A | C $\rightarrow$ D | D $\rightarrow$ C | B1 $\rightarrow$ B2 | B2 $\rightarrow$ B1 |
> | --- | --- | --- | --- | --- | --- | --- | --- | --- |
> | **PA** | 0.752 | 0.740 | $\underline{\text{0.817}}$ | 0.768 | 0.755 | 0.780 | 0.662 | 0.654 |
> | **GAA$_o$** | $\underline{\text{0.773}}$ | $\underline{\text{0.749}}$ | 0.815 | $\underline{\text{0.775}}$ | $\underline{\text{0.764}}$ | $\underline{\text{0.792}}$ | $\underline{\text{0.675}}$ | $\underline{\text{0.662}}$ |
> | GAA | **0.789** | **0.754** | **0.824** | **0.782** | **0.771** | **0.798** | **0.681** | **0.679** |
>
> **Table1-2.** Cross-network node classification on the citation, blog network.
>
> To further investigate the efficiency of **GAAo**, **Table 2** reports the running time comparison across various algorithms. We also compared the training time and GPU memory usage of common baselines UDAGCN [2] and a recent SOTA method, JHGDA, which align graph domain discrepancy hierarchical levels [3]. As shown in **Table2**, the evaluation results on **U $\rightarrow$ B** further demonstrate that, with tolerable computational and storage overhead, our method achieves superior performance.
>
> | Methods | Training Time (Normalized w.r.t. UDAGCN) | Memory Usage (Normalized w.r.t. UDAGCN) | Accuracy (%) |
> | --- | --- | --- | --- |
> | **UDAGCN** | 1   | 1   | 0.607 |
> | **JHGDA** | 1.314 | 1.414 | 0.695 |
> | **PA** | **0.498** | $\underline{\text{0.517}}$ | 0.679 |
> | **GAA$_o$** | $\underline{\text{0.504}}$ | **0.514** | $\underline{\text{0.697}}$ |
> | **GAA** | 1.063 | 1.113 | **0.704** |
>
> **Table2.** Comparison of Training Time, Memory Usage, and Accuracy on **U $\rightarrow$ B**.
>
> Following your suggestion, we also conduct one additional experiment on large-scale data MAG, and the resutls are shown in **Table 3**. It can be observed that GAA achieves state-of-the-art (SOTA) performance on large-scale datasets. Details of other baselines and dataset information are provided in the revised version (**Table 4** and **Table 1**).
>
> | Methods | US $\rightarrow$ CN | US $\rightarrow$ DE | US $\rightarrow$ JP | US $\rightarrow$ RU | US $\rightarrow$ FR | CN $\rightarrow$ US | CN $\rightarrow$ DE | CN $\rightarrow$ JP | CN $\rightarrow$ RU | CN $\rightarrow$ FR |
> | --- | --- | --- | --- | --- | --- | --- | --- | --- | --- | --- |
> | **PA** | 0.400 | 0.389 | 0.474 | $\underline{\text{0.371}}$ | 0.252 | $\underline{\text{0.452}}$ | 0.262 | 0.383 | 0.333 | 0.242 |
> | **GAA$_o$** | $\underline{\text{0.402}}$ | $\underline{\text{0.394}}$ | $\underline{\text{0.481}}$ | 0.363 | $\underline{\text{0.279}}$ | 0.445 | $\underline{\text{0.293}}$ | $\underline{\text{0.395}}$ | $\underline{\text{0.348}}$ | $\underline{\text{0.272}}$ |
> | GAA | **0.410** | **0.401** | **0.492** | **0.372** | **0.288** | **0.453** | **0.302** | **0.400** | **0.351** | **0.293** |
>
> **Table3.** Cross-network node classification on MAG datasets.
>
> [1] Liu, Shikun, et al. "Pairwise Alignment Improves Graph Domain Adaptation." *International Conference on Machine Learning*, 2024.
>
> [2] Man, Wu, et al. "Unsupervised Domain Adaptive Graph Convolutional Networks." *The Web Conference*, 2020.
>
> [3] Shi, Boshen, et al. "Improving Graph Domain Adaptation with Network Hierarchy." *Conference on Information and Knowledge Management*, 2023.

---

> > ### Comment · Reviewer_v8TX · 2024-11-23
> >
> > I appreciate the thorough rebuttal, and Appendix F addresses my concerns regarding the usefulness of the theoretical results. After seeing the comprehensive discussion in the rebuttal, I am now leaning toward accepting this paper. Therefore, I've raised the score to 6. Please ensure the efficient GAA results are also included in the manuscript.

---

> > > ### Author Response · Authors · 2024-11-24
> > > **Thank you for your feedback!**
> > >
> > > Dear Reviewer,
> > >
> > > Thank you for your thoughtful feedback and for increasing the score of our manuscript. We greatly appreciate your insightful questions, as addressing them has helped strengthen our paper. The efficient GAA results will be included in the revised submission, along with the other results. We welcome any further questions or suggestions you may have.
> > >
> > > Sincerely,
> > >
> > > The Authors

---

### Official Review · Reviewer_cb1H · 2024-11-09

**Soundness:** 3
**Presentation:** 4
**Contribution:** 3
**Rating:** 6
**Confidence:** 3

**Summary:**

This paper addresses the Graph Domain Adaptation (GDA) problem by investigating the critical role of node attributes in domain shift. This study reveals that node attributes contribute significantly to domain discrepancy. Through theoretical analysis using the PAC-Bayes framework, the authors prove that the domain shift is bounded by both topological structure differences and node attribute differences. Based on this insight, they propose Graph Attribute-driven Alignment (GAA), a novel method that aligns both structural and attribute information between domains. Experiments are conducted on two datasets: the Airpot network, Citation, Blog, and Social network.

**Strengths:**

- Unlike previous methods that primarily focused on graph structural information, this approach provides a novel perspective by emphasizing the importance of node attributes in GDA.
- A solid theoretical analysis proved that domain discrepancy comes from both graph structure and node attributes.
- Strong experimental results.

**Weaknesses:**

- The motivation for using an attention-like strategy is unclear.
- L_A contains both $||att^S - att^T||_2^2$ and $||att_f^S - att_f^T||_2^2$. There is no clear demonstration of which component is more important for performance. To highlight the effectiveness of aligning attributes, we need to check the effectiveness of $||att_f^S - att_f^T||_2^2$ (with/without Coss-view Matrix Refinement).
- The effect of $L_D$ is unclear. It is necessary to verify the contribution of $L_D$ compared to $L_A$, as $L_D$ is also a domain alignment strategy.

**Questions:**

- Please provide more information about the motivation behind the attention-like strategy (Eq. 8).
- How does the contribution of $||att_f^S - att_f^T||_2^2$ compare to $||att^S - att^T||_2^2$?
- Which is more important for the performance? $L_D$ or $L_A$?

---

> ### Author Response · Authors · 2024-11-23
> **Reply from the Authors**
>
> **Part 1/2**
>
> Thank you for your feedback. We would also appreciate your agreement on our method's novelty and effectiveness. Below, we address the concerns and questions raised in the weaknesses section. Please feel free to reach out if further clarification is required.
>
> ## Q1
>
> Our motivation is to make the attribute view matrix learnable. It selects essential attribute information to enable the model to obtain learnable attribute view information rather than relying on fixed attributes. This is essentially a **tradeoff between algorithm performance and computional complextity** to provide better performance.
>
> ## Q2
>
> The results from the $k$NN-GCN experiment (Tables 2 and 3) show that the Graph Convolutional Network (GCN) performs better when only relying on node attribute information. This finding highlights influence of attributes on model performance. We conducted ablation experiments focusing on this two specific terms: $||\textbf{att}^S - \textbf{att}^T||_2^2$ and $||\textbf{att}_f^S - \textbf{att}_f^T||_2^2$.
> The other loss functions used in model training are the same as those in GAA. Specifically, in the $\mathcal{L}_A$ function, **GAA$_T$** employs $||\textbf{att}^S - \textbf{att}^T||_2^2$ but removes $||\textbf{att}_f^S - \textbf{att}_f^T||_2^2$, while **GAA$_F$** utilizes $||\textbf{att}_f^S - \textbf{att}_f^T||_2^2$ but removes $||\textbf{att}^S - \textbf{att}^T||_2^2$. According to **Table1.** and **Table2.**, we observe that **GAA$_F$** generally outperforms **GAA$_T$**, suggesting that addressing attribute discrepancy is more beneficial for the performance of GDA. This is consistent with our observation that attributes discrepancy significantly larger than topology feature value discrepancy in our benchmarks. In addition, we discuss the impact of these two items separately in more detail in our revised manuscript **Appendix F**. In summary, while $||\textbf{att}_f^S - \textbf{att}_f^T||_2^2$ contributes more significantly to performance improvement, $||\textbf{att}^S - \textbf{att}^T||_2^2$ also plays a important role in enhancing the overall model.
>
> | Methods | U $\rightarrow$ B | U $\rightarrow$ E | B $\rightarrow$ U | B $\rightarrow$ E | E $\rightarrow$ U | E $\rightarrow$ B | DE $\rightarrow$ EN | EN $\rightarrow$ DE |
> | --- | --- | --- | --- | --- | --- | --- | --- | --- |
> | **GAA$_T$** | 0.672 | 0.544 | 0.526 | $\underline{\text{0.563}}$ | 0.534 | 0.678 | 0.762 | 0.738 |
> | **GAA$_F$** | $\underline{\text{0.686}}$ | $\underline{\text{0.549}}$ | $\underline{\text{0.532}}$ | 0.561 | $\underline{\text{0.536}}$ | $\underline{\text{0.686}}$ | $\underline{\text{0.769}}$ | $\underline{\text{0.744}}$ |
> | GAA | **0.704** | **0.563** | **0.542** | **0.573** | **0.546** | **0.691** | **0.779** | **0.751** |
>
> **Table1.** Cross-network node classification on the airport network and social network.
>
> | Methods | A $\rightarrow$ D | D $\rightarrow$ A | A $\rightarrow$ C | C $\rightarrow$ A | C $\rightarrow$ D | D $\rightarrow$ C | B1 $\rightarrow$ B2 | B2 $\rightarrow$ B1 |
> | --- | --- | --- | --- | --- | --- | --- | --- | --- |
> | **GAA$_T$** | 0.761 | 0.738 | 0.804 | $\underline{\text{0.776}}$ | 0.756 | 0.783 | 0.660 | 0.659 |
> | **GAA$_F$** | $\underline{\text{0.772}}$ | $\underline{\text{0.745}}$ | $\underline{\text{0.816}}$ | 0.771 | $\underline{\text{0.763}}$ | $\underline{\text{0.790}}$ | $\underline{\text{0.673}}$ | $\underline{\text{0.668}}$ |
> | GAA | **0.789** | **0.754** | **0.824** | **0.782** | **0.771** | **0.798** | **0.681** | **0.679** |
>
> **Table2.** Cross-network node classification on the Citation, Blog network.

---

> ### Author Response · Authors · 2024-11-23
> **Reply from the Authors**
>
> **Part 2/2**
>
> ## Q3
>
> $\mathcal{L}_D$ is a widely adopted unsupervised domain adaptation (UDA) loss in many graph domain adaptation (GDA) works [1, 2, 3], we include it to ensure a fair comparison. In **Section 5.5** ablation study of our manuscript, the variant **GAA_3** relies on the $\mathcal{L}_D$, which only leverages the topological information from the source and target graphs. Following your suggestion, we also conduct one additional experiment on the ablation experiment, including the performance of the model variant **GAA_D**, where $\mathcal{L}_D$ is excluded. We can observed that we observe that the model's performance is only slightly affected without $\mathcal{L}_D$, confirming that $\mathcal{L}_A$ plays a more important role.
>
> | Methods | U $\rightarrow$ B | U $\rightarrow$ E | B $\rightarrow$ U | B $\rightarrow$ E | E $\rightarrow$ U | E $\rightarrow$ B | DE $\rightarrow$ EN | EN $\rightarrow$ DE |
> | --- | --- | --- | --- | --- | --- | --- | --- | --- |
> | **GAA$_D$** | 0.696 | 0.557 | 0.538 | 0.567 | 0.539 | 0.682 | 0.772 | 0.744 |
> | GAA | **0.704** | **0.563** | **0.542** | **0.573** | **0.546** | **0.691** | **0.779** | **0.751** |
>
> **Table3.** Cross-network node classification on the airport network and social network.
>
> | Methods | A $\rightarrow$ D | D $\rightarrow$ A | A $\rightarrow$ C | C $\rightarrow$ A | C $\rightarrow$ D | D $\rightarrow$ C | B1 $\rightarrow$ B2 | B2 $\rightarrow$ B1 |
> | --- | --- | --- | --- | --- | --- | --- | --- | --- |
> | **GAA$_D$** | 0.782 | 0.747 | 0.820 | 0.778 | 0.768 | 0.786 | 0.674 | 0.671 |
> | GAA | **0.789** | **0.754** | **0.824** | **0.782** | **0.771** | **0.798** | **0.681** | **0.679** |
>
> **Table4.** Cross-network node classification on the Citation, Blog network.
>
> [1] Wu, Jun, et al. "Non-IID Transfer Learning on Graphs." *Annual AAAI Conference on Artificial Intelligence*, 2023
>
> [2] Man, Wu, et al. "Unsupervised Domain Adaptive Graph Convolutional Networks." *The Web Conference*, 2020.
>
> [3] Shi, Boshen, et al. "Improving Graph Domain Adaptation with Network Hierarchy." *Conference on Information and Knowledge Management*, 2023.

---

> > ### Comment · Reviewer_cb1H · 2024-11-25
> >
> > Dear authors,
> >
> > Thanks for your reply.
> >
> > It is noteworthy that both $GAA_T$ and $GAA_F$ perform well. This indicates that the proposed method is indeed superior to the baseline. However, the margin between $GAA_T$ and $GAA_F$ is also small, which makes it difficult to demonstrate that the node attribute-concentrated alignment is more effective than the topology-concentrated alignment. The performance of $GAA$ and $GAA_D$ is also similar, especially on the second dataset.
> >
> > I will keep my scores.

---

> > > ### Author Response · Authors · 2024-11-27
> > > **Reply from the Authors**
> > >
> > > **Part 1/2**
> > >
> > > Dear Reviewer,
> > >
> > > Thank you for your follow-up comments! We are happy to hear your thoughtful feedback and being supportive.
> > > There might be miscommuncations regarding the motivation of our work. We would like to note that **the focus of this paper is NOT to demonstrate that attribute-concentrated alignment is more effective than the topology-concentrated alignment.** Instead, the key insight of our work is  **both attribute and topology information plays crucial role for GDA**, while former has been long overlooked in existing methods.
> > >
> > > > + Firstly, by utilizing both attribute and topology of information, GAA performs better than the existing SOTA, as shown in Table 2, Table 3 and Table 4. In addition, to further validate the motivation of our work, we conducted an additional experiment to investigate the impact of attribute and topology information for GDA, respectively. Specifically, we andomly generated 1000 pairs of graphs (source and target) varying attribute and topology values. At each time, we first train a model using source values (either attribute or topology) and then evaluate its accuracy on the target graph (target accuracy). Figure 7 in Appendix F presents the normalized bound value divergence (calculated using Eq. (4) from Proposition 1) and loss value divergence $\mathcal{L}_A$ (defined in Eq. (12) on page 6) for both attribute and topology features as a function of value divergence. It can be observed that regardless of whether attributes or topology values are used, the target accuracy (green dots) consistently decreases as the divergence between the source and target domains increases. These empirical results further highlight that **both factors significantly influence the model's performance on the target domain**, while the importance of attributes has long been overlooked in existing GDA studies.
> > > >
> > > > + Secondly, the performance of $GAA_F$ and $GAA_T$ cannot reflect the impact of attribute or topology information -- only removing  $||\textbf{att}_f^S - \textbf{att}_f^T||_2^2$  (or $||\textbf{att}^S - \textbf{att}^T||_2^2$) cannot completely eliminate the effect of attribute (or topology) information in $GAA$, since both types of information intertwine to each other through **Cross-view Similarity Matrix Refinement (Eq. 9, Eq. 10)**.
> > > >
> > > > + Thirdly, in order to eliminate the impact of attribute information, we have conducted an ablation study in **Section 5.5**, where we construct **$GAA_3$ by removing the whole attribute graph channel** and only utilize the topology information through $\mathcal{L}_D$. **Table 5 and Table 6**  below compare the performances of $GAA$ and $GAA_3$, and it is evident that the accuracy of $GAA_3$ has dropped significantly, which further validates the rationale behind $GAA$.
> > > >
> > > > + Regarding $GAA_D$, **it is actually removing $\mathcal{L}_D$ rather than only utlizing $\mathcal{L}_D$**. The reason that removing $\mathcal{L}_D$ only slightly affects the performance is that **we already align topology information through $||\textbf{att}^S - \textbf{att}^T||_2^2$** (hence adding $\mathcal{L}_D$ does not provide additional performance gain). Here, we include $\mathcal{L}_D$ only for fair comparison purpose, since the other GDA algorithms also employ this term.
> > >
> > > In summary, our work emphasizes the importance of both topology and attribute for GDA, **which has not been proposed in previous work**. Moreover, our work also opens venues for future work to further leverage the attribute information for GDA.
> > >
> > > In the meanwhile, we really appreciate your question, which gives us the opportunity to further improve our manuscript. Specifically, in light of your comments, we have revised **Section 3 (line 203-208)** and **Section 6 (line 512-514)** to further clarify our motivation and avoid potential confusion.

---

> > > ### Author Response · Authors · 2024-11-27
> > > **Reply from the Authors**
> > >
> > > **Part 2/2**
> > >
> > > | Methods     | U $\rightarrow$ B | U $\rightarrow$ E | B $\rightarrow$ U | B $\rightarrow$ E | E $\rightarrow$ U | E $\rightarrow$ B | DE $\rightarrow$ EN | EN $\rightarrow$ DE |
> > > | ----------- | ----------------- | ----------------- | ----------------- | ----------------- | ----------------- | ----------------- | ------------------- | ------------------- |
> > > | **$GAA_3$** | 0.609             | 0.498             | 0.512             | 0.517             | 0.441             | 0.479             | 0.688               | 0.654               |
> > > | $GAA$       | **0.704**         | **0.563**         | **0.542**         | **0.573**         | **0.546**         | **0.691**         | **0.779**           | **0.751**           |
> > >
> > > **Table 5.** Cross-network node classification on the airport network and social network.
> > >
> > > | Methods     | A $\rightarrow$ D | D $\rightarrow$ A | A $\rightarrow$ C | C $\rightarrow$ A | C $\rightarrow$ D | D $\rightarrow$ C | B1 $\rightarrow$ B2 | B2 $\rightarrow$ B1 |
> > > | ----------- | ----------------- | ----------------- | ----------------- | ----------------- | ----------------- | ----------------- | ------------------- | ------------------- |
> > > | **$GAA_3$** | 0.685             | 0.631             | 0.731             | 0.681             | 0.719             | 0.655             | 0.623               | 0.611               |
> > > | $GAA$       | **0.789**         | **0.754**         | **0.824**         | **0.782**         | **0.771**         | **0.798**         | **0.681**           | **0.679**           |
> > >
> > > **Table 6.** Cross-network node classification on the Citation, Blog network.

---

### Meta-Review · Area_Chair_2Ygf · 2024-12-21

**Metareview:**

This paper focuses on addressing challenges in graph domain adaption, where labeled data in graph domains is often scarce. The authors highlight that most existing methods neglect the importance of node attributes and primarily focus on aligning structural features of graphs. They demonstrate that node attribute discrepancies significantly impact graph domain adaption performance, often more than topology shifts. They introduce Graph Attribute-Driven Alignment (GAA) algorithm, which incorporates both topology and attribute views for effective alignment. The experimental results demonstrate the effectiveness of the proposed method.

The overall quality of this paper is good. This paper has both theoretical and experimental analysis, which can which reveal that the difference of node attributes is more significant than the difference of topology structure and further emphasize the importance of node attribute alignment. The cross-channel module is proposed to align the graph topology and node attributes simultaneously, which is also demonstrated to be effective.

**Additional Comments On Reviewer Discussion:**

This paper finally receives the scores of 6, 6, 6, 6, which means, all the reviewers vote for weak acceptance. After the rebuttal process, the authors have addressed the key concerns raised from the reviewers, and thus all the reviewers finally gave a positive score to this paper. I would suggest that the authors could incorporate the suggestions and other minor issues pointed out by the reviewers into the final version of the paper.

---

### Decision · Program_Chairs · 2025-01-22

Accept (Poster)